# Ingroup sources enhance associative inference

Marius Boeltzig [1,2], Mikael Johansson [1] & Inês Bramão [1✉]

Episodic memory encompasses flexible processes that enable us to create and update knowledge by making novel inferences across overlapping but distinct events. Here we examined whether an ingroup source enhances the capacity to draw such inferences. In three studies with US-American samples ($N_{Study1} = 53$, $N_{Study2} = 68$, $N_{Study3} = 68$), we investigated the ability to make indirect associations, inferable from overlapping events, presented by ingroup or outgroup sources. Participants were better at making inferences based on events presented by ingroup compared to outgroup sources (Studies 1 and 3). When the sources did not form a team, the effect was not replicated (Study 2). Furthermore, we show that this ingroup advantage may be linked to differing source monitoring resources allocated to ingroup and outgroup sources. Altogether, our findings demonstrate that inferential processes are facilitated for ingroup information, potentially contributing to spreading biased information from ingroup sources into expanding knowledge networks, ultimately maintaining and strengthening polarized beliefs.

[1] Department of Psychology, Lund University, Lund, Sweden. [2] Department of Psychology, University of Münster, Münster, Germany. ✉email: ines.bramao@psy.lu.se

Our worlds are marked by social groups and inter-group dynamics. A preference for our group over others shapes perceptions and cognition[1]. Importantly, these ingroup biases lead to the formation of memories that align with the prevailing beliefs of the group. As individuals share and discuss their beliefs, memories become intertwined and shaped by the collective perspective[2]. This social influence can amplify polarization as group members validate and reinforce each other's partisan beliefs[3]. This study investigates the prediction that flexible memory processes, which support inferential decisions based on information presented across event boundaries, are shaped by social group influences. Inferential memory is crucial for knowledge creation, extension, and updating[4,5] and could therefore represent a factor contributing to the emergence and development of polarized beliefs.

Episodic memory allows us to mentally travel in time and re-experience events tied to particular times and places[6]. However, these memories are not static and veridical replays of our past. Instead, they are shaped by constructive processes that allow our ever-changing environment to be flexibly represented in our minds[7]. Critically, this malleable nature of memory may serve a socially adaptive function, allowing the content of our personal pasts to merge during social interactions to create a collective identity shared by individuals from the same social group[8].

Evidence for this idea comes from studies showing that people include details provided by others in descriptions of their own past[9], especially in interactions with people they like[10], trust[11], or that are part of their ingroup[12]. These findings can be explained by lowered source monitoring for the ingroup condition[13]. When the speaker is trusted, there is a lower need to assess the source of information, which increases the likelihood of misattributing the information provided by the speaker to the original event and incorporating it into the original memory[12].

Furthermore, social influences on memory have been described during learning processes. For instance, memory is better for material encoded in reference to us[14] and for material related to our social ingroup[15–17]. These findings demonstrate that our social world shapes memory processes, creating memory biases that may contribute to forming and perpetuating polarized beliefs. However, previous research has only focused on memory for discrete events. Thus, it remains unknown whether social group membership shapes flexible memory processes that extend beyond memory for single experiences.

Importantly, memory representations contain information not only acquired by direct experience but also inferred from multiple overlapping discrete events. For instance, if you meet a colleague drinking a coffee and later find a phone where she was sitting, you may infer that your colleague has lost her cell phone. This capacity to flexibly integrate details from our past to extract inferential associations across episodes is crucial for a wealth of cognitive processes, including spatial navigation, creativity, and learning[5,18]. Knowledge is built not only on isolated facts but also by integrating information[19], and previous studies have found that the ability to make inferences across event boundaries predicts academic success in children and young adults[20] above direct fact comprehension[21]. Notably, the process of making inferences across overlapping events has been shown in naturalistic settings such as the classroom[22], virtual museum exhibitions[23], and in the process of gathering medical information from several sources[24].

Given the ubiquitous nature of inferring relationships beyond those immediately perceived, it is crucial to understand whether this process is affected by biases driven by ingroup sources. Information provided by ingroup members is more likely to align with an individual's beliefs[25]. If this information is more likely to be assimilated and incorporated into their knowledge database, their initial beliefs are validated and reinforced, creating partisan minds with polarized beliefs.

Inferential associations across overlapping events are commonly investigated using the associative inference task[26,27]. In this task, participants encode associations between two elements (A and B) and later new associations involving one member of the previous pair (B and C). Participants are instructed to learn the indirect associations (A and C) by linking the two events through their overlapping shared element (B). Subsequently, an associative inference test assesses memory for the indirect link (AC) at retrieval. To test the prediction that an ingroup source facilitates the creation of inferential associations across overlapping events, we adapted this paradigm so that ingroup or outgroup sources presented the first paired associate (AB).

Across three studies ($N = 189$), we asked participants to encode AB associations (formed by an object and a background context depicting a location) presented by an ingroup or an outgroup source. AB encoding was followed by the encoding of overlapping BC associations (composed by a new object presented in the same context). At retrieval, participants were tested on the inferential AC associations; that is, they were asked to infer the indirect association between the two objects. The inferential association test was followed by a test for the source personas presenting each object. Finally, memory for the direct AB and BC associations was also tested.

The ingroup/outgroup manipulation was created by asking participants to compose source personas to be their teammates/opponents. Participants were asked to choose faces and create profiles that they liked for their teammates and that they disliked for their opponents (see Fig. 1). In all three studies, the subjective ratings of ease of encoding, as well as persona liking scores, confirmed the success of the group manipulation (see Supplementary Note 1).

Given that material presented by ingroup sources is better encoded and remembered[16], we predicted that AC associative inferences for material presented by ingroup sources would also be enhanced compared to inferences related to the material presented by outgroup sources. Additionally, we investigated source memory across successful and unsuccessful inferences for ingroup and outgroup information. As previous studies have found that successful inference leads to worse source memory[28–30] (even though this has recently been questioned[31]), we predicted lower source memory for successful compared to unsuccessful AC inferences. Moreover, because the outgroup increases source monitoring[12], we predicted worse source memory for information associated with the ingroup than the outgroup. We were agnostic as to how inference success and the social group would interact.

## Methods

**Ethics and inclusion statement**. The methods from all studies were conducted in accordance with the Swedish Act concerning the Ethical Review of Research involving Humans (2003:460) and the Code of Ethics of the World Medical Association (Declaration of Helsinki). As established by Swedish authorities and specified in the Swedish Act concerning the Ethical Review of Research involving Humans (2003:460), the present study does not require specific ethical review by the Swedish Ethical Review Authority due to the following reasons: (1) it does not deal with sensitive personal data, (2) it does not use methods that involve a physical intervention, (3) it does not use methods that pose a risk of mental or physical harm, (4) it does not study biological material taken from a living or dead human that can be traced back to that person.

In each study, we ensured that participants were fully informed about the types of data that would be collected, as well as the

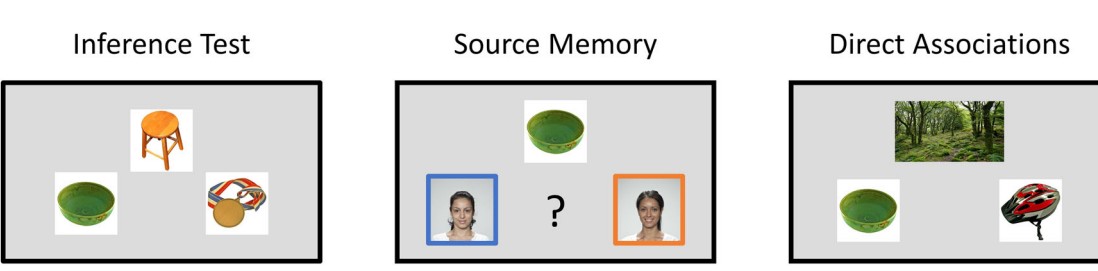

**a. Social Groups**

Ingroup  Outgroup

Political Views
Study Major
Eating Habits
...

I think that future interactions with these people would be pleasurable.

**b. Encoding**

AB encoding  BC encoding

**c. Memory Tests**

Inference Test  Source Memory  Direct Associations

**Fig. 1 Overview of the experimental paradigm. a** First, the social group manipulation was conducted, in which participants chose a face for each persona and assigned attributes from different categories to each team. In Studies 1 and 3, participants selected two personas per group; in Study 2, only one persona was included. A liking measure was administered as a manipulation check. Studies 1 and 2 used a relative scale where participants decided which team/persona each statement applied to more. In Study 3, the liking questionnaire was administered separately for ingroup and outgroup to obtain absolute liking ratings. **b** The encoding phase was divided into two separate encoding blocks, presenting the overlapping AB and BC associations, respectively. The backgrounds (B) were present in both pairs and were completed by an object each (A and C). One ingroup or outgroup persona presented each AB. In these episodes, the object was placed in a unique location on a circle, which served as the indicator for detail memory (detail memory was unaffected by the group manipulation and the results concerning this indicator are presented in Supplementary Note 3). After each encoding trial, participants were asked to rate how easy the display was to encode on a 3-point scale, where 1=easy and 3=hard. **c** In the subsequent memory tests, participants were prompted to make an associative inference by connecting the objects presented in the same context. They were also asked to indicate how sure they were about their choice on a 3-point scale, where 1=guessing, 2=maybe, and 3=sure. After testing all inferences in this manner, participants had to indicate by whom the objects were presented (source memory) and where on the screen they appeared (detail memory). Lastly, all direct associations were tested the same way as the inferences. The faces were selected from the Face Research Lab London Set[35] and the objects from the Bank Of Standardized Stimuli[33], both licensed under the Creative Commons Attribution-ShareAlike license (https://creativecommons.org/licenses/by/4.0/). The context pictures are similar to the ones used in the original experiment; however, for illustrative purposes we used license-free pictures from the unsplash data base (https://unsplash.com/license): the traffic jam picture by Iwona Castiello d'Antonio and the forest picture by Marc Pell.

usage, storage, and potential withdrawal procedures related to their data and consent. Participants were required to explicitly provide their consent to partake in the study by checking a designated box and entering their anonymous Prolific ID. Additionally, at the conclusion of the experiment, participants were debriefed regarding the objectives of the research. They were also given the possibility to contact the experimenter via e-mail or through the anonymous Prolific direct messages system. All participants were compensated following Prolific recommendations. The data collection was anonymous and did not involve recording any potentially identifying demographic information. Gender was assessed based on self-report by participants and was not mandatory to indicate. No data on ethnicity was collected.

Because we used the political party to prime participants with a group identity, we decided to test participants from the United States, where the two-party political system was expected to enhance the polarization between ingroup and outgroup identities compared to a multi-party system[32]. No researchers

from the United States took part in the investigation and given the compliance with Swedish laws and the Helsinki declaration, no local US ethical approval was sought.

**Participants: Study 1**. Previous studies using similar associative inference memory paradigms have shown reliable effects with sample sizes smaller than 30[28–30]. Additionally, within-subject design studies that investigate social influences on memory processes usually have sample sizes of around 50 participants[16,17]. As such, and to comply with previous relevant literature, we opened 60 slots on Prolific (www.prolific.co). Participants were recruited anonymously through the Prolific system and compensated in accordance with Prolific recommendations.

We excluded participants who did not produce incorrect inference trials as that precluded them from entering the planned analyses (n = 7). All the remaining participants met the inclusion criteria in that they responded to the encoding trials at least 90% of the time and reported liking the ingroup more than the outgroup personas at

the beginning of the experiment. The final sample consisted of 53 participants (40 female, 11 male, 2 diverse; mean age = 25.94, $SD = 2.12$, age range 22–30). Study 1 was not preregistered.

**Participants: Study 2.** As in Study 1, participants were recruited and paid via Prolific in a fully anonymous way. Participants from Study 1 were not eligible for participation in this experiment. This study was pre-registered (https://doi.org/10.17605/OSF.IO/QMZ32; 28 July 2022). To secure sufficient power to uncover the effects of interest, the sample size was determined in a power analysis based on the effect found in Study 1 by bootstrapping from the final sample of that study. For each sample size $n$ between 30 and 130, $n$ data points were randomly selected and entered a paired $t$-test contrasting ingroup and outgroup inference accuracy. This procedure was repeated 1000 times for each sample size and power was calculated as the frequency of significant results ($\alpha = 0.05$). A power of 80% was consistently reached with a sample of 68. We therefore decided to collect data until we had 68 participants who met the inclusion criteria. The final sample consisted of 23 women and 45 men (mean age = 26.70, $SD = 3.00$, age range 20–34).

**Participants: Study 3.** As in the other two studies, participants were recruited and paid via Prolific in a fully anonymous way. Study 3 was preregistered (https://doi.org/10.17605/OSF.IO/QMZ32; 28 July 2022). Based on the power analysis for Study 2, data were collected until a sample of $N = 68$ was reached (25 female, 38 male, 5 diverse; mean age = 24.82, $SD = 2.46$, age range 21–29). Participants were not eligible if they had taken part in our previous studies. Compliance with the previously applied inclusion criteria was monitored online, and participants were directly screened out if necessary and partially compensated. Additionally, one participant who failed to recognize their teammates and opponents at the end of the experiment was excluded.

**Materials: Study 1.** We constructed 64 ABC-triplets by selecting 128 objects from the BOSS database[33] and 64 context background scenes from one of our previous experiments[34]. Each triplet comprised two objects (A and C) and a scene (B) with no apparent pre-experimental associations. The triplets were divided into two lists, which were counter-balanced across participants to ingroup and outgroup, respectively, to ensure that differences between conditions were not due to the stimulus material.

Additionally, participants selected their ingroup and outgroup persona from eight female faces obtained from the Face Research Lab London Set[35]. Only female faces were used to avoid possible interactions with participants' gender. We attempted to select faces that could evoke different reactions in different participants by choosing those with higher standard deviations in the attractiveness ratings. Furthermore, the personas' profiles comprised six categories (political views, major, eating habits, favorite sports, hobbies, and favorite music). Participants could select one of five possible attributes for each category to assign to their ingroup and outgroup personas. The categories and attributes were chosen in accordance with stereotypes about political orientations[36].

**Material: Study 2.** The same triplets from Study 1 were used for this study and stimuli were counter-balanced across participants.

**Material: Study 3.** A random selection of 36 triplets from the previous studies was used. The material was divided into two lists, each assigned to either the ingroup or the outgroup. Lists were counter-balanced across participants, ensuring that differences between conditions were not due to stimulus material.

**Procedure: Study 1.** Data collection was conducted online and consisted of a Qualtrics questionnaire and the main memory task run on Pavlovia[37]. First, participants gave informed consent and provided minimal demographics. Next, to prime participants with their political identities to influence the construction of the personas' profiles, participants were asked about their party affiliation and political ideology and completed a 14-item questionnaire assessing the identification with their partisanship[38].

Next, participants were redirected to Pavlovia and were asked to construct the ingroup and the outgroup personas. Participants first chose two faces they liked, which would serve as their ingroup personas, referred to as teammates, and two dislikable faces to constitute their outgroup, referred to as opponents. Participants also chose team colors, which were displayed as a frame around the persona pictures throughout the experiment. Next, participants were presented with six profile categories, for each of which they could choose one of five attributes for their ingroup and one for their outgroup. They were asked to produce profiles typical of a teammate and an opponent. To anchor profile construction in political identity, this was always the first category, and participants chose between strong or leaning Democrat or Republican as well as Independent—the other five categories followed in the same manner. More rounded profiles were preferred over sole political labels for increased ecological validity, strengthened liking and disliking, and a chance for less political participants to construct meaningful ingroups and outgroups.

After constructing the profiles, participants were again shown the full profiles and asked to imagine a real encounter with each persona. To evaluate whether the manipulation successfully induced liking for the teammate, participants filled out an interpersonal liking measure with five items[39]. This measure covers perceptions about future interactions (e.g., "I would like to get to know these people better") and cognitive evaluations about the person (e.g., "I think that these people and I may have a lot in common.") The measure was administered with a relative seven-point scale, where each of the teams formed one pole of the scale. Participants were asked to judge which of the two teams each statement applied to more. Values lower than the scale's midpoint indicated a relative outgroup preference, and values above the midpoint reflected an ingroup preference.

After the persona construction, participants were trained on the task. To ensure high data quality in the less controlled online setting, participants whose performance indicated guessing and a lack of understanding of the instructions had the chance to repeat the training. If they still showed performance below chance afterwards, they were screened out and compensated for their time.

The main task comprised two blocks, each consisting of an AB encoding block, followed by a block of BC, a distractor task, and memory tests. First, participants were presented with 32 AB associations. Each trial started with a fixation cross for 1 s, followed by the background scene (B) with the persona presented in the center for 2 s. After that, the object (A) was added and remained on the screen with the scene and the persona for four more seconds. Each A-object appeared at a unique position on a circle drawn around the persona. Subsequently, the 32 BC associations were presented. Each trial started with a fixation cross, presented for 1 s, followed by the background context (B) with a superimposed object (C) for 4 s. Participants were instructed to memorize the scenes, objects, locations, and personas and were given explicit instructions to connect the

two objects presented in the same scene. After each AB and BC encoding trial, participants rated the ease of encoding on a 3-point scale, where 1 indicated *easy* and 3 *hard* to encode (see Fig. 1). To participants, the encoding phase was introduced as a trip through a series of places (B) where they were also asked to imagine actually being. For the AB associations, they were told that either one of their teammates or opponents would be waiting for them in each place and present an object (A).

After a 20 s mathematical distractor task, participants were tested on the material in three steps. First, AC inferences were tested, followed by memory for detail and source and memory for the direct associations (BC and AB). All 32 AC inferences were tested by cueing participants with the A object and presenting the correct C and a lure that the same persona had indirectly presented. Participants chose the object with a key press and afterwards rated how confident they were in their response using a 3-point scale, where 1 indicated *guessing* and 3 *sure*. After the inference test, detail and source memory for all As was tested. For each object, detail memory was tested by asking participants to indicate the object's original position on the screen using the mouse. However, detail memory was not influenced by social group and did not interact with inference success. Therefore, the results of this indicator are reported in the Supplementary Material (see Supplementary Note 3). Next, source memory was tested by asking participants to choose the persona that had acted as the source of the object. All four personas were on screen and could be chosen by the participant. To reduce noise in the data, originating from random choices, an additional option *I don't remember* was available to the participants in the source test. The direct recognition memory test followed. BC associations were tested first, followed by the ABs, using the same procedure as in the inference test. Each test decision had to be made within a 6 s limit (see Fig. 1).

After the memory task, the liking measure was repeated. Lastly, participants read a full debrief and had the chance to report disturbances and distractions during the experiment.

**Procedure: Study 2**. The procedure of this study was identical to Study 1 with the sole exception that there was only one source persona per group. Participants therefore only chose one teammate and one opponent from the set of eight faces and constructed profiles for the individual personas rather than the team. Consequently, there were only two personas to choose from in the source memory test, along with the "I don't know" option.

**Procedure: Study 3**. The procedure was identical to Study 1 with some adjustments. Instead of using one relative scale for persona liking, we asked participants to judge the items from the previously used IL-6[39] first for the ingroup personas and then for the outgroup personas, which provided separate scores for ingroup and outgroup liking. The material was presented in a single experimental block. For alignment with previous studies[28–30], the explicit instructions to encode the detail memory (i.e., position of the objects on the screen) was removed. Instead, participants were told that the objects would appear at a random location. After the inference and source tests, participants were asked to retrieve the locations of the objects in a surprise test (see Supplementary Note 3). Furthermore, at the end of the experiment, participants were asked to identify their teammates and opponents from the set of the original eight faces from the profile construction as a quality check. Only one participant was excluded from data analysis based on this criterion.

**Data analysis: Study 1**. All data was analyzed using R[40]. Data distribution was assumed to be normal, but this was not formally tested. First, we checked if the group manipulation worked by

looking into the liking and the subjective ease of encoding ratings. The liking measure comprised five items that were averaged for each participant. A one-sample *t* test was used to contrast the liking rating with the zero midpoint. A difference in this test indicated a preference for either the ingroup or the outgroup persona. Additionally, we contrasted the subjective ease of encoding ratings across groups using a dependent sample *t* test (see Supplementary Note 1).

To further assess the effects of the group manipulation, we used a partial source memory indicator[41] to investigate if participants were more likely to make source memory errors within a team. A tendency to do so would indicate a stronger perception of the personas as a team. Previous research has shown that outgroups are more strongly organized as a homogenous team, while ingroup members are more highly individuated[42,43]. The results of this analysis, reported in Supplementary Note 4, corroborate previous research, and indicate that our group manipulation had consequences consistent with those previously reported.

To investigate the effect of social group on making novel associative inferences across overlapping events, we assessed performance on the inference task and memory for detail and source. First, performance for novel inferences (AC) and direct associations (AB, BC) was contrasted across social group with a repeated-measures ANOVA including the factors group (ingroup vs. outgroup) and association type (AC vs. AB vs. BC). Although not the focus of this study, differences between associations were followed up with post-hoc tests using Tukey correction. Additionally, ingroup and outgroup performance was contrasted within each association type in pre-planned direct contrasts using two-tailed paired *t* tests.

The analysis was performed on accuracy, response times, and confidence ratings. Even though the BCs were not directly associated with a social identity, the persona can come to mind during BC encoding and/or during the inference test and can then be attached to them. Accuracy for each participant was calculated as the percentage of correct responses in the association test per condition. Trials for which participants did not respond within the time limit were excluded from the analysis ($M = 0.54\%$ per participant). To dampen the effects of potential outliers, the response times were analyzed using the median for each condition and participant. The confidence ratings were analyzed using the mean for each condition and participant. Only correct recognition trials were included in the response times and confidence ratings analyses. An exploratory analysis tested whether any accuracy ingroup biases on the ACs could be predicted by a potential ingroup biases on the direct associations. To that end, ingroup bias scores were computed for each participant by subtracting outgroup performance from ingroup performance, with higher values indicating a stronger ingroup bias on accuracies.

Next, we investigated if social group and inference success affected source memory. Performance on source memory was assessed with a repeated-measures ANOVA with the factors group (ingroup vs. outgroup) and performance on the inference test (correct vs. incorrect). Using two source personas per group allowed us to utilize two conceptually different source memory indicators. Memory for the persona only considered responses as correct when the correct persona was chosen, while memory for the team more leniently also accepted responses as correct if the other persona from the correct group was chosen. Direct pre-planned contrasts were carried out with two-tailed paired *t* tests to investigate patterns specific to ingroup and outgroup. Accuracy for source was calculated as the percentage of trials for which participants indicated the correct source persona. Trials where participants did not respond or responded "I don't remember"

were excluded from the analysis ($M = 9.49\%$ trials per participant). Importantly, there was no significant evidence for group differences in these responses ($t(52) = 0.61$, $p = 0.547$, $d = 0.08$, 95%CI[−0.19, 0.36]). All ANOVAs are also summarized in Supplementary Note 5 and all descriptive values can be inspected in Supplementary Note 6.

Any null results that were critical to our interpretations were corroborated within a Bayesian framework. While frequentist statistics can only lead to a failure to reject the null hypothesis, Bayesian analyses can quantify the support in its favor. All $t$-tests were conducted in JASP[44] as two-sided tests with a Cauchy prior of $r = 0.707$, which is the default in JASP, and pitted the null hypothesis, stating no difference between two conditions, against the alternative hypothesis, which assumed an effect size different from zero. Furthermore, Bayes Factors (BF) were reported when the $p$-values fell between 0.05 and 0.10, where they are sometimes described as marginally significant and for exploratory correlation analyses (using the default stretched beta prior width $\kappa = 1$). Each Bayes Factor is reported together with the Median of the posterior distribution ($Md_{posterior}$) and its 95% credible interval (95% $CI_{posterior}$). The sensitivity of Bayes Factors to different priors was assessed visually using the JASP robustness check and deemed robust in each reported case.

Additionally, to further assess source memory, we tested how it was affected by ease of encoding ratings (Supplementary Note 2). The results support the idea that trusted ingroup information is less likely to be tagged with a source when encoding is challenging. In contrast, outgroup information will more likely be labeled as untrustworthy, even when encoding is difficult.

**Data analysis: Study 2**. The strategy for data analysis was the same as in Study 1. Note, however, that in this study, the source memory analysis was only conducted for persona. Study 2 used only one persona per team and therefore we cannot distinguish source memory for persona from source memory for team. Again, trials with no answer in the memory tests were excluded from the analysis ($M = 0.75\%$ per participant), and response times and confidence were analyzed using only correct trials. Responses corresponding to "I don't remember" and non-responses ($M = 11.03\%$) were omitted for the source analysis. As in Study 1, there was no evidence for a significant difference between groups in these responses ($t(67) = 0.05$, $p = 0.960$, $d = 0.01$, 95% CI[−0.23, 0.25]).

**Data analysis: Study 3**. The data were analyzed in the same way as in the previous studies. Trials with no response on the final association test ($M = 0.44\%$) were excluded from the analysis, and response times and confidence were analyzed using only correct trials. For the source memory analysis, we excluded "I don't remember" and missing responses ($M = 7.68\%$). There was no statistically significant difference between ingroup and outgroup in the amount of excluded responses ($t(67) = 0.49$, $p = 0.626$, $d = 0.06$, 95%CI[−0.18, 0.30]).

**Reporting summary**. Further information on research design is available in the Nature Portfolio Reporting Summary linked to this article.

## Results

In all three studies, both the liking measures and subjective ease of encoding responses showed clear ingroup bias, indicating that the persona manipulation was successful (see Supplementary Note 1).

**Study 1**. For this study, we collected data from an online sample of 53 US-Americans (see Methods for details). To assess the effect of ingroup source on inferential and direct associations, a repeated-measures ANOVA with the factors association type (AB, BC, AC) and group (ingroup, outgroup) was conducted for accuracy as well as response times and confidence ratings of correct responses. An effect of association type was observed for accuracy ($F(2,104) = 110.33$, $p < 0.001$, $\eta^2 = 0.31$), response times ($F(2,104) = 305.16$, $p < 0.001$, $\eta^2 = 0.52$) and confidence ($F(2,104) = 72.61$, $p < 0.001$, $\eta^2 = 0.18$), reflecting performance differences between direct and inferential associations. Direct associations were made more accurately (ABs: $t(52) = 10.49$, $p < 0.001$, $d = 1.44$, 95%CI[1.05, 1.83]; BCs: $t(52) = 14.16$, $p < 0.001$, $d = 1.95$, 95%CI[1.48, 2.41]), faster (ABs: $t(52) = 19.20$, $p < 0.001$, $d = 2.64$, 95%CI[2.06, 3.21]; BCs: $t(52) = 18.52$, $p < 0.001$, $d = 2.54$, 95%CI[1.98, 3.10]), and more confidently (ABs: $t(52) = 8.78$, $p < 0.001$, $d = 1.21$, 95%CI[0.85, 1.56]; BCs: $t(52) = 11.52$, $p < 0.001$, $d = 1.58$, 95%CI[1.17, 1.99]) than AC inferences. BC associations were also retrieved significantly more accurately than ABs ($t(52) = 4.30$, $p = 0.002$, $d = 0.59$, 95% CI[0.30, 0.89]) but neither significantly faster ($t(52) = 1.08$, $p = 0.862$, $d = 0.15$, 95%CI[−0.12, 0.42]) nor more confidently ($t(52) = 2.95$, $p = 0.112$, $d = 0.41$, 95%CI[0.12, 0.69]).

We also observed a significant effect of group on accuracies ($F(1,52) = 8.16$, $p = 0.006$, $\eta^2 = 0.01$) and confidence ratings ($F(1,52) = 11.80$, $p = 0.001$, $\eta^2 = 0.01$), reflecting a general ingroup advantage. This was not apparent on response times ($F(1,52) = 1.84$, $p = 0.184$, $\eta^2 = 0.001$). There were no significant interactions (accuracy: $F(2,104) = 2.06$, $p = 0.132$, $\eta^2 = .005$; response times: $F(2,104) = 0.05$, $p = 0.951$, $\eta^2 = 0$; confidence: $F(2,104) = 2.16$, $p = 0.120$, $\eta^2 = 0.002$).

Consistent with our prediction that inferential associations would be enhanced for information provided by ingroup members, the direct contrast between ingroup and outgroup showed a clear ingroup advantage for ACs in terms of accuracy ($t(52) = 2.52$, $p = 0.015$, $d = 0.35$, 95%CI[0.07, 0.63]) and confidence ($t(52) = 2.90$, $p = 0.006$, $d = 0.40$, 95%CI[0.11, 0.68]), but not for response times ($t(52) = 0.44$, $p = 0.660$, $d = 0.06$, 95% CI[−0.21, 0.33]). Additionally, an ingroup advantage was observed for the direct ABs in terms of accuracy ($t(52) = 2.12$, $p = 0.038$, $d = 0.29$, 95%CI[0.01, 0.57]) and for BCs in terms of confidence ($t(52) = 3.14$, $p = 0.003$, $d = 0.43$, 95%CI[0.15, 0.72]; see Fig. 2).

An exploratory analysis showed that the ingroup advantage observed on the ABs did not significantly correlate with the ingroup advantage observed on the ACs ($r = 0.03$, 95% CI[−0.25, 0.29], $p = 0.852$, $BF_{01} = 5.74$, $Md_{posterior} = 0.03$, 95% $CI_{posterior}$[−0.24, 0.29]). There is therefore no support for the possibility that the ingroup advantage on associative inferences could be explained by the ingroup advantage observed in direct associations.

Next, to explore the mechanisms that could drive a boost in inferential decisions for material provided by ingroup sources, we tested how source memory differed between ingroup and outgroup and how it was affected by inference success, using repeated-measures ANOVAs with those two factors. The use of two source personas per group permitted utilizing two source memory indicators, namely memory for persona (i.e., correctly choosing the persona that presented the object) and a coarser memory for the team, where source judgments were also accepted as correct when participants chose the wrong persona from the correct team. First, contrary to the previous literature[28–30] but consistent with a recent study[31], we observed a significant effect of inference success both for memory for persona ($F(1,52) = 10.36$, $p = 0.002$, $\eta^2 = 0.03$) and team ($F(1,52) = 10.64$, $p = 0.002$, $\eta^2 = 0.04$), reflecting better source memory for

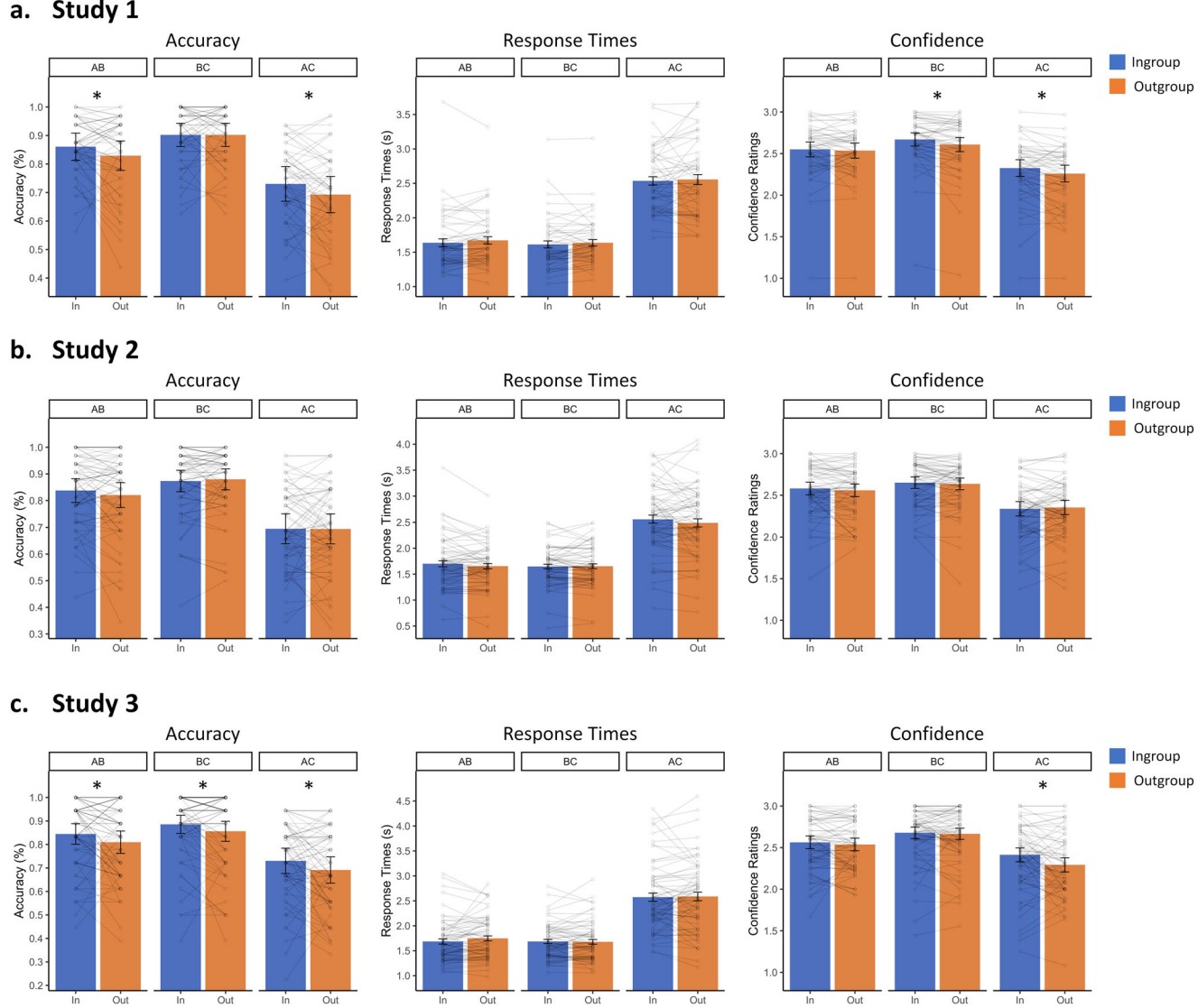

**Fig. 2 Accuracy, Response Times, and Confidence for Studies 1-3.** Plotted is the average memory performance indicated by accuracies, response times, and confidence ratings for a. Study 1, b. Study 2, and c. Study 3. The mean average for each participant is also shown. The error bars represent the standard error of the mean (SE; Study 1: $n = 53$, Studies 2 & 3: $n = 68$). Highlighted (*) are the comparisons showing a significant group effect ($p < 0.05$).

correct compared with incorrect inferences. Second, an effect of group was observed for team memory ($F_{(1,52)} = 8.90$, $p = 0.004$, $\eta^2 = 0.03$), revealing the predicted higher source memory in the outgroup. On persona memory, the effect of group gained no significance ($F_{(1,52)} = 0$, $p = 0.946$, $\eta^2 = 0$) and there was no interaction ($F_{(1,52)} = 1.07$, $p = 0.305$, $\eta^2 = 0.002$).

Interestingly, there was a significant interaction between the two factors on memory for the team ($F_{(1,52)} = 6.65$, $p = 0.013$, $\eta^2 = 0.01$). While outgroup team memory was not significantly different between correct and incorrect inferences ($t(52) = 1.25$, $p = 0.218$, $d = 0.17$, 95%CI[−0.10, 0.45], $BF_{01} = 3.22$, $Md_{posterior} = 0.16$, 95%CI$_{posterior}$[−0.10, 0.43]), ingroup team memory was better for correct than incorrect inferences ($t(52) = 3.82$, $p < 0.001$, $d = 0.52$, 95%CI[0.23, 0.81]). Additionally, outgroup team memory was better than ingroup team memory for incorrect inferences ($t(52) = 3.27$, $p = 0.002$, $d = 0.45$, 95%CI[0.16, 0.73]), while this difference did not gain significance for correct inferences ($t(52) = 0.99$, $p = 0.327$, $d = 0.14$, 95%CI[−0.14, 0.41], $BF_{01} = 4.21$, $Md_{posterior} = 0.13$, 95%CI$_{posterior}$[−0.13, 0.39]; see Fig. 3).

This pattern of results suggests that source monitoring resources are differentially allocated to ingroup and outgroup sources. Previous studies have shown that outgroup sources are generally judged as unreliable[45] and may therefore be essential to encode in order to use the information with caution in the future. Accordingly, participants may preferentially allocate their attentional resources to the outgroup source and do not encode the information they provide. Consequently, they fail to make inferences based on outgroup information, even though they keep the source of those specific events. On the other hand, trustworthy ingroup information requires less source monitoring[12], and the encoding of the source is therefore not as important. Consequently, participants may still be capable of making inferences based on ingroup information but without remembering the source of the information.

A differential allocation of attentional resources when encoding information from ingroup and outgroup sources should be most pronounced under challenging encoding conditions, where participants have to decide what to prioritize for encoding[46]. When encoding is easy, no such prioritization has to be made,

## a.  Study 1

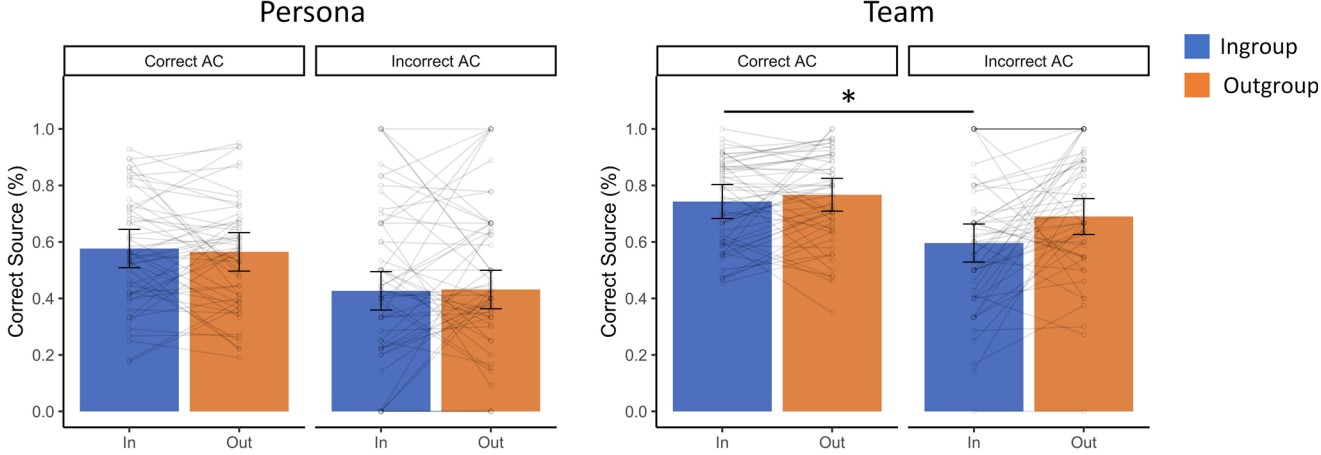

## b.  Study 2

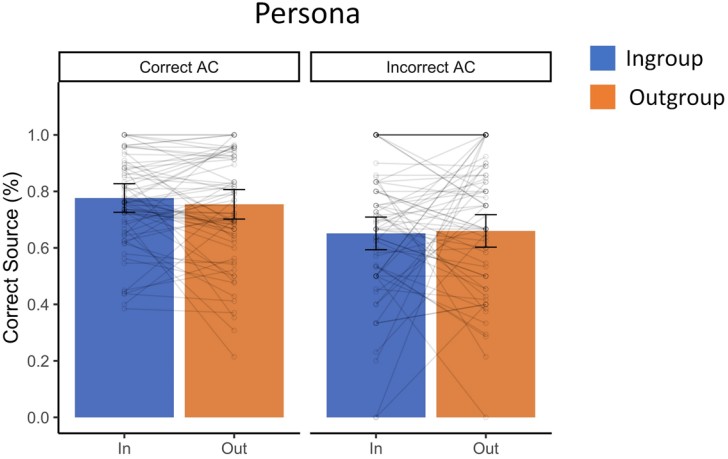

## c.  Study 3

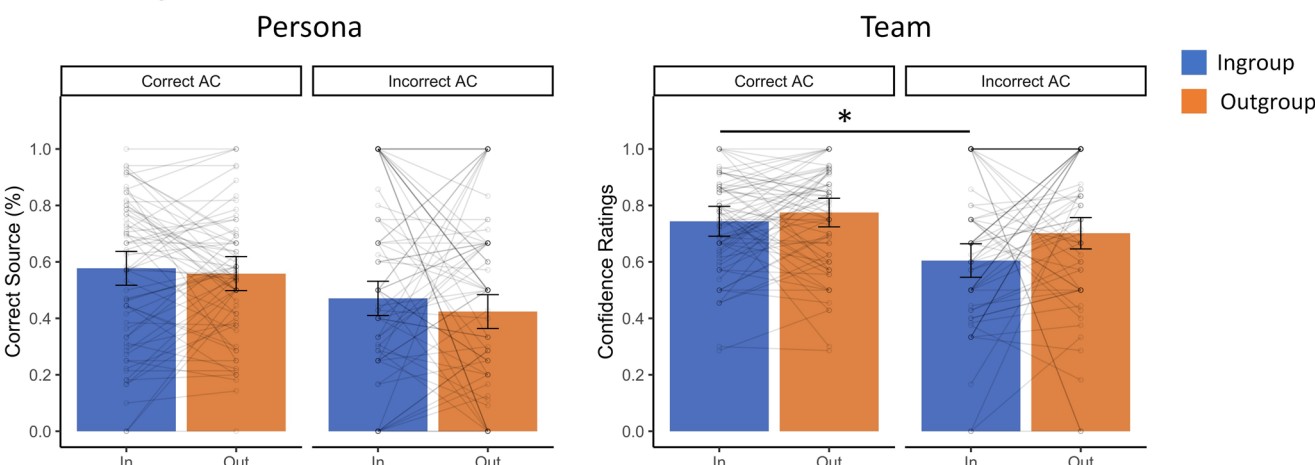

**Fig. 3 Source Memory for Studies 1-3.** Plotted is the average source memory for (**a**) Study 1, (**b**) Study 2, and (**c**) Study 3. The mean average for each participant is also shown. The error bars represent standard error of the mean (*SE*; Study 1: $n = 53$, Studies 2 & 3: $n = 68$). Highlighted (*) are the comparisons showing a significant effect ($p < 0.05$).

allowing comprehensive encoding of the whole episode. To validate this, we investigated how team memory for ingroup and outgroup varied as a function of the subjective ease of encoding, when AC inferences were accurate (see Supplementary Note 2). Our results indicate that, during trials with correct AC inferences, ingroup source memory diminishes when participants perceive encoding as challenging. However, outgroup source memory remains consistent irrespective of the ease of encoding. This finding corroborates the view that in demanding encoding situations, participants can still derive inferences based on ingroup information, even if memory for the ingroup source of information fades.

**Study 2**. In this pre-registered study (https://doi.org/10.17605/OSF.IO/QMZ32; 28 July 2022), we aimed to replicate the findings of Study 1. The procedure was identical to Study 1, except that the information was presented by only one source persona per group. This allowed us to test whether the source personas had to be part of a group or whether liked individuals could also produce the ingroup advantage. Previous studies investigating social influences on memory, such as social contagion, have found the social effects not to be limited to ingroups, but to generalize to known, trusted, or liked individuals[12,47,48]. Given the implications that this extension of the effect would have, this study investigates if the ingroup bias effect found in Study 1 generalizes from groups to liked individuals. Based on a power analysis (see Methods for a detailed description), we recruited 68 US-Americans to participate in this online study.

Our findings showed a significant effect of association type on accuracies ($F(2,134) = 139.04$, $p < 0.001$, $\eta^2 = 0.24$), response times ($F(2,134) = 206.24$, $p < 0.001$, $\eta^2 = 0.41$), and confidence ratings ($F(2,134) = 60.47$, $p < 0.001$, $\eta^2 = 0.17$). As in Study 1, direct associations were made more accurately (ABs: $t(67) = 13.86$, $p < 0.001$, $d = 1.68$, 95%CI[1.31, 2.05]; BCs: $t(67) = 14.07$, $p < 0.001$, $d = 1.71$, 95%CI[1.33, 2.08]), faster (ABs: $t(67) = 15.30$, $p < 0.001$, $d = 1.86$, 95%CI[1.46, 2.25]; BCs: $t(67) = 15.60$, $p < 0.001$, $d = 1.89$, 95%CI[1.49, 2.29]), and more confidently (ABs: $t(67) = 8.55$, $p < 0.001$, $d = 1.04$, 95%CI[0.74, 1.33]; BCs: $t(67) = 10.09$, $p < 0.001$, $d = 1.22$, 95%CI[0.91, 1.54]) than inferences. Additionally, BC associations were retrieved significantly better than ABs ($t(67) = 4.25$, $p = 0.014$, $d = 0.52$, 95%CI[0.26, 0.77]) while there was no evidence for an advantage in response times ($t(67) = 0.88$, $p = 0.890$, $d = 0.11$, 95%CI[−0.13, 0.35]) or confidence ($t(67) = 2.53$, $p = 0.092$, $d = 0.31$, 95%CI[0.06, 0.55]). However, there were no effects of group (accuracy: $F(1,67) = 0.34$, $p = 0.564$, $\eta^2 = 0$; response times: $F(1,67) = 3.08$, $p = 0.084$, $\eta^2 = 0.001$; confidence: $F(1,67) = 0.28$, $p = 0.600$, $\eta^2 = 0$) and no interactions (accuracy: $F(2,134) = 1.12$, $p = 0.329$, $\eta^2 = 0.001$; response times: $F(2,134) = 1.48$, $p = 0.232$, $\eta^2 = 0.001$; confidence: $F(2,134) = 1.15$, $p = 0.320$, $\eta^2 = 0.001$).

The critical comparisons between ingroup and outgroup ACs on accuracy ($t(67) = -0.04$, $p = 0.969$, $d = 0$, 95%CI[−0.24, 0.24], $BF_{01} = 7.33$, $Md_{posterior} = -0.03$, 95%CI$_{posterior}$[−0.26, 0.21]) and confidence ($t(67) = -0.63$, $p = 0.532$, $d = -0.08$, 95%CI[−0.32, 0.16], $BF_{01} = 6.21$, $Md_{posterior} = -0.07$, 95%CI$_{posterior}$[−0.31, 0.16]), which had gained significance in Study 1, did not indicate an ingroup advantage in this study (see Fig. 2). As before, there was also no significant difference on response times ($t(67) = 1.52$, $p = 0.134$, $d = 0.18$, 95%CI[−0.06, 0.43], $BF_{01} = 3.59$, $Md_{posterior} = 0.14$, 95%CI$_{posterior}$[−0.09, 0.38]).

Additionally, source memory for persona (the use of only one persona did not allow the investigation of memory for team) showed a significant effect of inference success ($F(1,67) = 7.90$, $p = 0.006$, $\eta^2 = 0.02$), revealing better source memory for correct compared with incorrect inferences. However, no effect of group ($F(1,67) = 0.02$, $p = 0.877$, $\eta^2 = 0$) and no interaction between

inference success and group ($F(1,67) = 2.41$, $p = 0.125$, $\eta^2 = 0.004$) were observed (see Fig. 3).

In sum, Study 2 showed no effects of group on inferential AC associations or source memory. Importantly, using only one persona per group increased the amount of information linked to each persona. This could have created mnemonic competition between the associations related to each persona, contributing to memory interference[49,50]. If this had been the case, however, we should have observed a decrease in general memory performance in Study 2 compared with Study 1. However, this is not the case, and the performance for Study 2 was comparable to Study 1.

Alternatively, the lack of support for an ingroup advantage for inferences in Study 2 suggests that the ingroup advantage reported in Study 1 might depend on the perception of a group and is not evident with liked individuals. In fact, previous studies have shown that group perception increases with the number of personas added to the group[51]. Moreover, a preliminary experiment from our lab with only one persona per group but presenting the BC associations (instead of the ABs) did not show evidence for an inference advantage for information presented by an ingroup source either (see Supplementary Note 7).

**Study 3**. Study 1 found an advantage on ingroup inferences when the ingroup and outgroup consisted of two personas each, while Study 2 found no such effect with only one persona per group. This pattern may be related to perceiving the sources as part of groups[51]. To test the robustness of the finding from Study 1, we conducted a pre-registered (https://doi.org/10.17605/OSF.IO/QMZ32; 28 July 2022) replication study in an independent sample of 68 US-Americans that followed the procedures employed in Study 1 with two improvements. To ensure that participants remembered the personas who belonged to their ingroup/outgroup throughout the whole task, we included a persona recognition test at the end of the experiment. Additionally, instead of using a relative liking scale as depicted in Fig. 1, we opted to assess liking for ingroup and outgroup separately for two reasons. First, this provided a more stringent manipulation check as it allowed us to show that the ingroup was not merely preferred over the outgroup, but also liked (with values above the midpoint of the scale) on its own merits (see Supplementary Note 1). Second, participants were now able to rate ingroup and outgroup liking separately without having to compare personas with each other and make a relative judgement. With this improved and less taxing way of assessment, we wanted to test whether the ingroup preference co-varied with the AC ingroup advantage. Such a demonstration would more convincingly tie the effect to social group.

The results showed a significant effect of association type on accuracies ($F(2,134) = 68.08$, $p < 0.001$, $\eta^2 = 0.16$), response times ($F(2,134) = 151.31$, $p < 0.001$, $\eta^2 = 0.40$), and confidence ratings ($F(2,134) = 50.47$, $p < 0.001$, $\eta^2 = 0.14$). Inferences were less accurate (ABs: $t(67) = 8.51$, $p < 0.001$, $d = 1.03$, 95%CI[0.73, 1.33]; BCs: $t(67) = 10.58$, $p < 0.001$, $d = 1.28$, 95%CI[0.96, 1.61]), slower (ABs: $t(67) = 13.80$, $p < 0.001$, $d = 1.67$, 95%CI[1.30, 2.05]; BCs: $t(67) = 12.42$, $p < 0.001$, $d = 1.51$, 95%CI[1.16, 1.86]), and less confident (ABs: $t(67) = 6.83$, $p < 0.001$, $d = 0.83$, 95%CI[0.55, 1.11]; BCs: $t(67) = 10.12$, $p < 0.001$, $d = 1.23$, 95%CI[0.91, 1.55]) than direct associations. Furthermore, participants were more confident ($t(67) = 3.14$, $p = 0.022$, $d = 0.38$, 95%CI[0.13, 0.63]) for BCs than ABs, but there was no evidence for an advantage on response times ($t(67) = 1.02$, $p = 0.844$, $d = 0.12$, 95%CI[−0.12, 0.36]) or accuracy ($t(67) = 3.19$, $p = 0.052$, $d = 0.39$, 95%CI[0.14, 0.64]). Additionally, an effect of group, signaling an ingroup advantage across association types, was evident for accuracy ($F(1,67) = 11.49$, $p = 0.001$, $\eta^2 = 0.01$), and confidence ($F(1,67) = 6.14$, $p = 0.016$,

$\eta^2 = 0.004$), but not for response times ($F(1,67) = 0.81$, $p = 0.371$, $\eta^2 = 0.001$).

Crucially, we replicated the results of Study 1 by observing an ingroup advantage on accuracies for AC inferences ($t(67) = 2.18$, $p = 0.033$, $d = 0.26$, 95%CI[0.02, 0.51]). Once again, there was no significant group difference on response times ($t(67) = 0.30$, $p = 0.762$, $d = 0.04$, 95%CI[−0.20, 0.28]). There were also ingroup advantages for the accuracy of ABs ($t(67) = 2.13$, $p = 0.037$, $d = 0.26$, 95%CI[0.01, 0.50]) and BCs ($t(67) = 2.20$, $p = 0.031$, $d = 0.27$, 95%CI[0.02, 0.51]; see Fig. 2). Additionally, we found a significant interaction between group and association type on confidences ($F(2,134) = 3.29$, $p = 0.040$, $\eta^2 = 0.004$), revealing a significant ingroup advantage on ACs ($t(67) = 2.90$, $p = 0.005$, $d = 0.35$, 95%CI[0.10, 0.60]), but not on ABs ($t(67) = 0.62$, $p = 0.536$, $d = 0.08$, 95%CI[−0.16, 0.32], $BF_{01} = 6.24$, $Md_{posterior} = 0.07$, 95%CI$_{posterior}$[−0.16, 0.30]) or BCs ($t(67) = 0.62$, $p = 0.536$, $d = 0.08$, 95%CI[−0.16, 0.32], $BF_{01} = 6.24$, $Md_{posterior} = 0.07$, 95%CI$_{posterior}$[−0.16, 0.30]). There were no interactions for accuracy ($F(2,134) = 0.10$, $p = 0.905$, $\eta^2 = 0$) or response times ($F(2,134) = 1.21$, $p = 0.302$, $\eta^2 = 0.001$).

Next, to determine if the ingroup bias observed in inferential memory could be attributed to the ingroup bias in the direct associations, we correlated these effects across participants in an exploratory analysis. Consistent with Study 1, there was no correlation between AB and AC ingroup biases ($r = 0.09$, 95% CI[−0.16, 0.32], $p = 0.483$, $BF_{01} = 5.19$, $Md_{posterior} = 0.08$, 95% CI$_{posterior}$[−0.15, 0.31]). Additionally, the BC ingroup advantage, which was significant in this study, did not correlate with the AC advantage either ($r = 0.17$, 95%CI[−0.07, 0.39], $p = 0.164$, $BF_{01} = 2.56$, $Md_{posterior} = 0.16$, 95%CI$_{posterior}$[−0.07, 0.39]). This suggests that the direct ingroup advantages are not the only factors driving the inference bias effect.

The source memory analysis showed no significant effects for memory for persona (inference success: $F(1,67) = 1.60$, $p = 0.210$, $\eta^2 = 0.003$; group: $F(1,67) = 1.11$, $p = 0.295$, $\eta^2 = 0.003$; interaction: $F(1,67) = 0.48$, $p = 0.489$, $\eta^2 = 0.001$), while source memory for the team likened the findings reported in Study 1. There was no effect of inference success ($F(1,67) = 2.98$, $p = 0.089$, $\eta^2 = 0.01$), but there was a significant effect of group ($F(1,67) = 4.15$, $p = 0.046$, $\eta^2 = 0.01$). Despite a non-significant interaction ($F(1,67) = 1.18$, $p = 0.282$, $\eta^2 = 0.003$), the direct contrasts revealed a similar pattern as in Study 1. That is, we observed no statistical difference between correct and incorrect inferences in the outgroup team memory ($t(67) = 0.52$. $p = 0.602$, $d = 0.06$, 95%CI[−0.18, 0.30], $BF_{01} = 6.58$, $Md_{posterior} = 0.06$,

95%CI$_{posterior}$[−0.17, 0.29]), but ingroup team memory was lower for incorrect compared to correct inferences ($t(67) = 2.04$, $p = 0.045$, $d = 0.25$, 95%CI[0.004, 0.49]). Moreover, there was a tendency for source memory to be lower for the ingroup compared with the outgroup for incorrect inferences, which was however not supported within a Bayesian approach ($t(67) = 1.81$, $p = 0.075$, $d = 0.22$, 95%CI[−0.02, 0.46], $BF_{10} = 0.62$, $Md_{posterior} = 0.21$, 95%CI$_{posterior}$[−0.03, 0.45]). Team memory was not significantly different between ingroup and outgroup for correct inferences ($t(67) = -1.00$, $p = 0.322$, $d = -0.12$, 95%CI[−0.36, 0.12], $BF_{01} = 4.67$, $Md_{posterior} = -0.12$, 95%CI$_{posterior}$[−0.35, 0.12]; see Fig. 3c). Again, this pattern suggests that source monitoring resources are differentially allocated to ingroup and outgroup. Source memory for the outgroup is higher, even when inference fails. This suggests that attentional resources are allocated to monitoring the outgroup source to the detriment of the information provided by it. This explanation is corroborated by a further analysis investigating source memory as a function of the ease of encoding (Supplementary Note 2).

Additionally, an unregistered exploratory analysis used the improved liking measure and assessed whether the ingroup inference advantage correlated with ingroup preference. To that end, each participant's outgroup AC accuracy was subtracted from their ingroup AC accuracy to obtain an individual measure of advantage magnitude, and outgroup liking was subtracted from ingroup liking to form an estimate of ingroup liking bias. When considering the whole sample, there was no significant correlation between pre-experimental ($r = 0.22$, 95%CI[−0.02, 0.43], $p = 0.075$) or post-experimental ($r = 0.17$, 95%CI[−0.08, 0.39], $p = 0.177$) ingroup liking bias and the AC inference advantage. However, when excluding participants whose source memory for the team was numerically below chance (<50%; $n = 9$), pre-experimental ($r = 0.27$, 95%CI[0.02, 0.49], $p = 0.037$) and post-experimental liking bias ($r = 0.28$, 95%CI[0.03, 0.50], $p = 0.031$) significantly correlated with the AC ingroup advantage. This suggests that the ingroup advantage is higher for those participants that created stronger ingroup/outgroup liking differences and therefore strongly ties the inference advantage to the social group (see Fig. 4).

In sum, Study 3 replicated the finding that inferences across overlapping episodes are performed more accurately and confidently when an ingroup source presents the information. An exploratory analysis showed that this ingroup bias co-varies with ingroup liking measures, clearly tying this finding to the social group. Critically, the

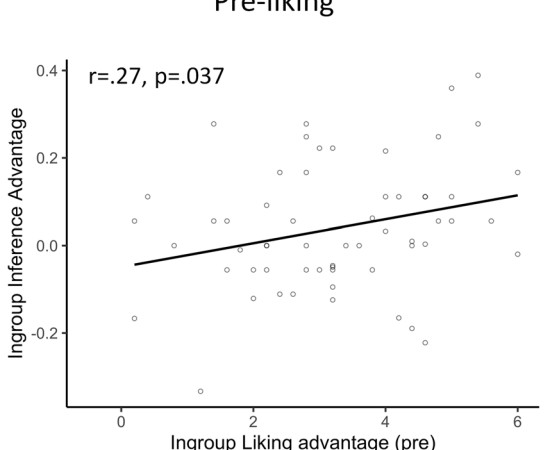
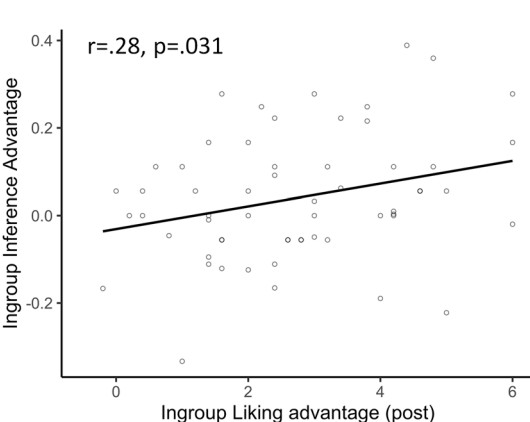

**Fig. 4 Correlation between the inference advantage and persona liking.** This analysis includes the participants whose source memory performance for the team was above chance ($n = 59$).

source memory findings offer an insight into the mechanisms behind this ingroup advantage. Our data suggest that trusted ingroup sources do not require extensive monitoring resources during encoding. As such, participants can focus on encoding the information provided by the ingroup personas and therefore are more effective at making inferences based on this information.

## Discussion

This study reports the first evidence that an ingroup source facilitates associative inference across separate episodes. This finding significantly extends previous literature on ingroup advantages in memory[16] by showing enhanced inferential memory for ingroup information. Flexibly recombining information from overlapping events and thus going beyond what is immediately observed within the neat boundaries of discrete events is crucial to updating existing knowledge structures and building new concepts and representations[4,5,20].

People are highly motivated to create and maintain shared realities within their ingroups[25,52], which critically includes a shared view of the past[53,54]. Information that we get from ingroup members is consequently more likely to align with our own views of the world. Our research indicates that people tend to rely heavily on information pertaining to their own group when making inferences across overlapping events. This affects how knowledge is created and updated[55,56], and may reinforce polarized beliefs. Our study is the first to demonstrate that ingroup bias extends beyond distinct events and may impact broader knowledge networks.

Interestingly, Study 3 showed that participants with stronger ingroup inference biases were the ones who also showed a stronger ingroup bias in the liking measures. This demonstration clearly ties our findings to the social group and may have important societal implications.

For instance, you may read that an organization is announcing a fundraiser (A) to clean up the local park (B). A couple of weeks later, when you are taking a walk in the park, you may be pleasantly surprised about its improved state (C). Our results suggest that you are more likely to attribute the cleanliness of the park to the fundraiser (i.e., make the AC inference) if it would have been announced by an organization you like or are part of than if the fundraiser would have been arranged by a group you dislike. Partisan divides would be protected by a reduced ability to make inferences across overlapping but temporally distinct experiences.

The data further suggest that the AC ingroup advantage may be tied to varying source monitoring resources allocated to ingroup versus outgroup information. Outgroup source memory remained consistent across correct and incorrect inference trials. Moreover, when inference was correct, outgroup source memory did not vary based on the perceived ease of encoding, while ingroup source memory declined in more difficult encoding conditions (Supplementary Note 2). Together, these findings suggest that a differential prioritization mechanism is at play when encoding is more challenging. There seems to be a consistent emphasis on encoding outgroup sources, even under demanding conditions. This may stem from the potential necessity to use outgroup information cautiously in future situations. In difficult encoding conditions, this heightened focus on the outgroup might compromise the encoding of the event itself, leading to diminished mnemonic flexibility for outgroup information. This is not the case for trustworthy ingroup information; when encoding is difficult, the encoding of the source may be secondary to the episode itself. As a result, participants can derive inferences from ingroup information even if the source of the information is lost. The relationship between source memory and inference success implies that source monitoring resources are

an important mechanism when accounting for how inferences vary across event boundaries for ingroup and outgroup information.

Our data also showed that no ingroup advantage for inference performance was observed when only one persona per group was used (Study 2). This suggests that this advantage might be dependent on the perception of group[51]. However, this needs to be taken with caution as previously reported mnemonic ingroup advantages have often been found with liked individuals as well[10]. In Study 2, the high number of associations connected with the two personas may have created a fan effect that may have impeded performance. Future work could introduce several liked and disliked personas that do not form part of a group to test the generalizability of the finding.

Successful inferences may stem from integrating past and current information when encoding new events[57]. In this context, when encoding BC, the related AB association is retrieved, leading to a unified ABC representation that aids in inferring the AC relationship. Simultaneously, successful inferences can also be made on demand during the AC inference test, by flexibly retrieving and recombining two past episodes[58]. The enhanced encoding of ingroup information[15,16] may have boosted both of these mechanisms. Interestingly, we found that ingroup advantages for direct associations did not predict the inference ingroup advantage. This implies that the ingroup advantage for inferences goes beyond enhanced encoding for ingroup information. Future studies should investigate how and when group affiliation mediates inferential memory processes. Given that group bias can arise from ingroup favoritism and/or outgroup derogation[59], future work should also determine if ingroup cues enhance inferences above a neutral baseline or if outgroup cues impede them.

**Limitations**. Future work may also test the generalizability of the ingroup inference effect we found here. The choices of holistic profiles, self-constructed by participants, with several attributes for each group, and the US-American sample, were motivated by maximizing group distinctions in a natural setting but may leave open questions about other group inductions and samples. Future studies may instead choose to use minimal group paradigms[60] or single pre-existing social identities. Another avenue for future work would be to develop and include neutral baselines to assess the relative contributions of ingroup and outgroup information on associative inference performance. Lastly, participants completed the study online without supervision, which may introduce more noise to the data than traditional laboratory experiments. However, this should not introduce a systematic bias for ingroup or outgroup memory.

## Conclusions

In sum, our results extend previous literature by providing the first demonstration that ingroup sources augment the capacity to make novel inferences from separate but overlapping episodes. This process is crucially involved in updating and building new knowledge representations. Strikingly, this finding was observed using randomly generated episodes, suggesting that even neutral information from an unfavored source enjoys less privileged access to our knowledge base, leading to partisan minds. In real-life situations where information can be rejected and doubted or elicit strong reactions, the inferential memory effects are likely to be even more pronounced.

## Data availability
The data collected in the three studies and presented in this manuscript, aggregated on the level of each participant, are openly available on Open Science Framework (https://osf.io/n3f9p/?view_only=0c97eba9c9264c18afa1d3faefcff01d).

## Code availability

The code used for analysis is available on Open Science Framework (https://osf.io/n3f9p/?view_only=0c97eba9c9264c18afa1d3faefcff01d).

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

## Acknowledgements
This work was supported by the Swedish Research Council Grant VR 2019-02455. The funders had no role in study design, data collection and analysis, decision to publish or preparation of the manuscript. We thank all volunteers who participated in this study.

## Author contributions
M.B. contributed to conceptualization, methodology, investigation, formal analysis, data interpretation, writing – original draft, writing – review & editing, and visualization. M.J. contributed to conceptualization, methodology, data interpretation, and writing (review & editing). I.B. contributed to conceptualization, methodology, formal analysis, data interpretation, writing (review & editing), visualization, supervision, and funding acquisition.

## Funding

## Competing interests
The authors declare no competing interests.
