## [Peer Review File · Communications Psychology]

20th Jun 23

Dear Dr Bramao,

Thank you for submitting your manuscript titled "Inferential memory is moderated by the ingroup/outgroup status of the information source" to Communications Psychology. We have given the paper our careful consideration and find it of potential interest. However, due to certain shortcomings we are concerned that sending the current manuscript out to review could lead to unnecessary delays and quite possibly an undesirable outcome of the review process.

In particular, the manuscript contains multiple statements about results that are central to the interpretation that are not accompanied by appropriate statistics; for example, in the results section that describe Study 2, you write, "However, and in contrast with Study 1, no effects of group were observed neither in the inferential nor in the direct associations (see Fig. 2)". It is journal policy to require positive evidence for the null, i.e. all statements about no difference/no effect/no association must be backed up by appropriate statistics, such as Bayesian statistics (BF must yield more than anecdotal evidence for the null) or equivalence tests. Please provide appropriate statistics -in the text- for all statements that describe the absence of an effect where this absence forms the basis of interpretation.

We shall hope to receive your revised version as soon as you are able to complete the suggested revisions. If something similar is published in the interim we will have to consider the impact it has on the novelty of a revised manuscript.

If you anticipate a delay of more than four weeks, please let us know. Should your manuscript be substantially delayed without notifying us in advance and your article is eventually published, the received date may be that of the revised, not the original, version.

If you are not interested in submitting a suitably revised manuscript in the future please let me know immediately so we can close your file. If you have any questions, please contact me.

Please use the link below when you are prepared to resubmit.

[link redacted]

Thank you for your interest in Communications Psychology.

Best regards,
Marike

Marike Schiffer, PhD
Chief Editor
Communications Psychology

18th Aug 23

Dear Dr Bramao,

Thank you for your patience during the peer-review process. Your manuscript titled "Inferential memory is moderated by the ingroup/outgroup status of the information source" has now been seen by 2 reviewers, and I include their comments at the end of this message. They find your work of interest, but raised some important points. We are interested in the possibility of publishing your study in *Communications Psychology*, but would like to consider your responses to these concerns and assess a revised manuscript before we make a final decision on publication.

We therefore invite you to revise and resubmit your manuscript, along with a point-by-point response to the reviewers. Please highlight all changes in the manuscript text file.

Editorially, we note that both reviewers are asking for more justification about key experimental decisions including sample size, the motivation for each experiment, and various analytical choices (e.g., how is team source memory specified? what was the rationale for looking at partial source memory?). Reviewer #2 provides guidance to steer away from calling non-significant results "marginal" and instead provide Bayes factors for what you deem to be "marginal" effects. These requests are in line with the journal's requirements for statistics reporting and interpretation as detailed in the checklist and template linked below (under EDITORIAL POLICIES AND FORMATTING).

To facilitate processing your revised manuscript, we ask you to work closely with this checklist to ensure that your revision complies with all our requirements. To mention a few outstanding issues:

- Please include a Data Availability Statement and a Code Availability statement compliant with our guidelines. We strongly recommend you already deposit the code (and ideally data), as this may otherwise delay the potential acceptance of your manuscript at a later stage. The code deposition should utilize version control and doi-mint the code.
- Please state explicitly whether the local guidelines rule the studies exempt from an IRB-type ethics approval.

Please use the following link to submit your revised manuscript, point-by-point response to the referees' comments (which should be in a separate document to any cover letter) and the completed checklist:

[link redacted]

We hope to receive your revised paper within 8 weeks; please let us know if you aren't able to submit it within this time so that we can discuss how best to proceed. If we don't hear from you, and

the revision process takes significantly longer, we may close your file. In this event, we will still be happy to reconsider your paper at a later date, provided it still presents a significant contribution to the literature at that stage.

Please do not hesitate to contact me if you have any questions or would like to discuss these revisions further. We look forward to seeing the revised manuscript and thank you for the opportunity to review your work.

Best regards,

Jesse Rissman

Jesse Rissman, PhD
Editorial Board Member
Communications Psychology
orcid.org/0000-0001-8889-5539

EDITORIAL POLICIES AND FORMATTING

Editorial Policy: [Policy requirements](https://www.nature.com/documents/nr-editorial-policy-checklist.pdf) (Download the link to your computer as a PDF.)

Furthermore, please align your manuscript with our format requirements, which are summarized on the following checklist:

[Communications Psychology formatting checklist](https://www.nature.com/documents/commspsychol-style-formatting-checklist-article-rr.pdf)

and also in our style and formatting guide [Communications Psychology formatting guide](https://www.nature.com/documents/commspsychol-style-formatting-guide-accept.pdf) .

*** TRANSPARENT PEER REVIEW:** Communications Psychology uses a transparent peer review system. This means that we publish the editorial decision letters including Reviewers' comments to the authors and the author rebuttal letters online as a supplementary peer review file. However, on author request, confidential information and data can be removed from the published reviewer reports and rebuttal letters prior to publication. If your manuscript has been previously reviewed at another journal, those Reviewers' comments would not form part of the published peer review file.

* **CODE AVAILABILITY:** All Communications Psychology manuscripts must include a section titled "Code Availability" at the end of the methods section. In the event of publication, we require that the custom analysis code supporting your conclusions is made available in a publicly accessible repository; at publication, we ask you to choose a repository that provides a DOI for the code; the link to the repository and the DOI will need to be included in the Code Availability statement. Publication as Supplementary Information will not suffice. We ask you to prepare code at this stage, to avoid delays later on in the process.

* **DATA AVAILABILITY:**

All Communications Psychology manuscripts must include a section titled "Data Availability" at the end of the Methods section or main text (if no Methods). More information on this policy, is available at <http://www.nature.com/authors/policies/data/data-availability-statements-data-citations.pdf>.

At a minimum the Data availability statement must explain how the data can be obtained and whether there are any restrictions on data sharing. Communications Psychology strongly endorses open sharing of data. If you do make your data openly available, please include in the statement:

We recommend submitting the data to discipline-specific, community-recognized repositories, where possible and a list of recommended repositories is provided at <http://www.nature.com/sdata/policies/repositories>.

If a community resource is unavailable, data can be submitted to generalist repositories such as [figshare](https://figshare.com/) or [Dryad Digital Repository](http://datadryad.org/). Please provide a unique identifier for the data (for example a DOI or a permanent URL) in the data availability statement, if possible. If the repository does not provide identifiers, we encourage authors to supply the search terms that will return the data. For data that have been obtained from publicly available sources, please provide a URL and the specific data product name in the data availability statement. Data with a DOI should be further cited in the methods reference section.

REVIEWERS' EXPERTISE:

Reviewer #1 episodic memory; associative inference
Reviewer #2 episodic memory; group membership

REVIEWERS' COMMENTS:

Reviewer #1 (Remarks to the Author):

The study investigated how memory for object-scene pairs (AB) is affected by ingroup/outgroup status of a “persona” presented alongside them, as well as how the ingroup/outgroup status affects the likelihood of episodic inference that connects those AB associations with overlapping BC associations (AB & BC, therefore AC). The results showed that participants better remember object-scene AB associations and better inferred AC relationships when the AB associations were “presented” by an ingroup persona than outgroup persona. There was a flip side to this effect as the source memory (memory who presented a given object) was worse for ingroup than outgroup source. These findings provide an interesting connection between previously separate lines of research on episodic inference with research on source identity memory effects, and allude to how our biases towards information from ingroup sources may affect not only memory for individual information but also our tendency to connect related memories in service of knowledge formation.

This was a refreshing paper to read and a review to write. There was a lot to like. The studies were well designed, two studies were pre-registered, the authors self-replicated Study 1 in Study 3 after not finding an effect with a different manipulation in Study 2. The analyses were appropriate, results were interesting, the conclusions were well supported and carefully worded. Finally, the whole paper was very well written. It seems like a lot of thought went into each analysis decision as well as how much and where to report (e.g., adding Bayesian statistics when appropriate, including follow up exploratory analyses to corroborate some of the results).

I only have a few points I would like to see addressed.

1. Team source memory? One important set of findings related to team source memory, but it’s not possible to evaluate those as written. There needs to be more clarity and details for the methods, especially how it’s even possible to get the “team” correct but persona incorrect in a 2-alternative forced-choice test. As written and visualized in Fig 1, there were only two personas to choose from on every trial. Presumably, one of them was correct but the nature of the foil is not explained. If it’s always a persona from another team, then it wouldn’t be possible to choose a wrong persona from a correct team. If the foil can be from the same team as the correct choice (on some percentage of trials?), then essentially both answers are correct when analyzing for “team” source correct score, so it’s again not possible to compare 100% sure correct options across conditions, unless “?” (don’t know) responses are counted as incorrect on those trials. Either way, this needs to be much clearer.

2. AB effect driving AC effect? With one exception, all AC differences were accompanied by AB differences. It would be important to know if the AB differences fully account for AC differences or if AC differences go above and beyond AB differences. Are people unable to make AC inference because they don’t remember the AB trials? Or are there additional inference effects even after AB’s were taken into account. There is probably many ways how to address this question. For instance, Zeithamova and Preston 2017 did two control analyses when faced with the same issue of AC differences being potentially driven by AB differences, one looking at across-subjects correlations, one limiting AC analysis to only those trials where AB was remembered. Just to clarify, the paper is still a worthwhile reading, even if it turns out that the AC differences can be fully explained by AB differences.

3. In the discussion, the authors provide an example that concludes: “Our results suggest that you

would be less likely to attribute this to the fundraiser of the disliked group than if your own group had been responsible.” I was confused about this claim and did not find it consistent with the data. All the source memory effects (for team) were driven by lower Ingroup Incorrect Inference trials, with the remaining conditions (outgroup correct and incorrect AC, ingroup correct AC) being all about equal. Either revise the example or provide a more thorough handholding through the logic of your claim.

4. Detail memory. Detail memory was delegated to the supplement because no effects were found, which is fine. However, given the Carpenter & Schacter vs. de Araujo Sanches & Zeithamova conflicting findings, the null effect may be actually interesting. I would definitely like to see more information, even if it remains in the supplement. At minimum, please report the actual data with the actual error magnitude for each condition (not just anova results). I also didn't understand the scaling to values between 0 and 1 and how that would help with large variability, unless it's a nonlinear conversion. Nevertheless, using analyses on each subject's medians instead of means seemed like an appropriate choice given that the error distributions are necessarily very skewed.

Minor comments:

- Fig 1 legend: “In Study 3, the items were assessed ..”. Maybe switch to “the personas” to avoid confusion with objects?
- Why were the subjects US-Americans when the research team isn't?
- Provide age range in addition to the mean age for participants in all studies

Reviewer #2 (Remarks to the Author):

This manuscript investigated the effects of ingroup vs. outgroup status of information source on memory for inferential associations in three studies. In all studies, participants were first shown an object picture on a background image (AB) that was presented by a persona whose views/interests were similar vs. dissimilar to their own (ingroup vs. outgroup), and then a different object picture on the same background image (BC). Later, participants' memory for indirect, associative inferences (AC), direct associations (AB, BC), as well as for the information source (source memory: whether an object was presented by ingroup or outgroup) and the location (detail memory) of the individual objects were probed. Studies 1 and 3 used two personas per group, whereas Study 2 had only one persona per group. In Studies 1 and 2, interpersonal liking as a manipulation check was measured once on a relative scale (with the ingroup at one end of the scale and the outgroup at the other), whereas in Study 3 it was measured separately per each group. In all studies, participants liked ingroup personas more than outgroup personas. Also consistent across the studies, memory was better for direct AB, BC associations than for associative AC associations. Of direct relevance to this manuscript's inquiry, in Studies 1 and 3, there was an ingroup advantage in associative AC inference (as well as in memory for AB and BC associations). Also, in Study 1, source memory was better following successful than unsuccessful associative inference, but only for ingroup trials but not for outgroup trials, with outgroup source memory significantly better than ingroup source memory following successful inference. These source memory results were largely replicated in Experiment 3, with a caveat that some of the comparisons did not reach statistical significance. In comparison to these results from Studies 1 and 3, in Study 2, the authors found no significant effect of group on associative AC inference, direct AB and BC associations, or source memory.

This is an interesting paper that addresses an empirical question of how social group membership influences memory integration. It is a nice extension of previous studies on the effects of social group membership on memory processes, and the findings of this study will be informative for future studies aimed at investigating specific proximal mechanisms through which the perception of ingroup vs. outgroup affects everyday memory processes. The studies appear to have been competently designed and conducted, the statistical analyses are appropriate to the research question, and the findings are clearly reported. Overall, I found the manuscript well-organized and well-written. However, I have a number of suggestions for improving the manuscript that I list below, roughly in the order they appear in the manuscript.

The authors describe that their sample size for Study 1 was based on the sample size in previous studies but these studies they mention appear to bear no relevance to the critical ingroup vs. outgroup manipulation. Furthermore, the authors make no mention at all of power or expected effect sizes. Did the authors consult any past studies that looked at the effects of social group on memory processes (e.g., Jeon et al., 2021; Marsh, 2020 that the authors themselves cite in the manuscript)? Was the sample size of Study 1 sufficient to detect ingroup-related memory effects reported in such studies? In fact, given that this manuscript is concerned with memory for indirect, inferential associations rather than direct associations, the authors should expect their effect sizes to be somewhat smaller than the effect sizes reported in previous memory studies. In any case, I suggest that the authors offer an improved rationale for the sample size for Study 1.

There appears to be some room for improvement in terms of describing questions motivating Studies 2 and 3. For example, in Study 2, the authors briefly mention that they wanted to see if it is liked individuals vs. ingroup that gives rises to the ingroup advantage in associative inference, but provide no description at all about relevant background literature and/or hypotheses. Likewise, in Study 3, the authors describe that they chose to measure interpersonal liking separately for each group (rather than once using a relative scale as in Studies 1 and 2) in order to examine a potential correlation between the ingroup advantage in associative inference and ingroup liking, but to me it is not very clear why the relative scale could not be used to test for the correlation. I suggest the authors to further elaborate on the motivation/rationale for each study to improve their already well-written manuscript even further.

Across both the main text and the supplementary material, there are a good number of analyses that the authors performed, but without providing clear justification (e.g., categorizing trials based on the ease of encoding for source memory analysis, partial source memory analysis). For example, it is not very clear why the authors decided to look at partial source memory, and how the results of partial source memory relate to ingroup vs. outgroup inferential success and/or to the absence of ingroup advantage in associative inference when there is only one persona (in Study 2). I suggest the authors to provide a clearer justification/explanation for why they made certain analytical choices and discuss the results of different analyses in light of the main inquiry of the manuscript. This will greatly help the reader to more clearly follow the results from each separate analysis and to better understand their implications.

For the sake of completeness, I would like to ask the authors to report Bayes factors (i.e., “amount of evidence”) for not only null effects but also significant effects consistently throughout the manuscript. On a related note, I suggest the authors to steer away from calling non-significant results as “marginal” and provide Bayes factors for what they deemed “marginal”. Doing so will

allow the reader to gauge the amount of evidence for the reported non-significant results (and any follow-up planned comparisons that were done despite the non-significant null effect).

In the discussion, the authors state “The higher source monitoring resources allocated to the outgroup may interfere with encoding the episode itself, leading to lowered mnemonic flexibility for outgroup information. This is different for trustworthy ingroup information where the encoding of the source may be secondary to the episode itself.” (on p. 20). To me this is difficult to follow: if what the authors suggest is the case, shouldn't source memory for outgroup trials be better following incorrect than correct AC inferences (i.e., incorrect AC inferences resulting from even greater interference with encoding the episode itself due to outgroup source monitoring)? Likewise, if encoding of the ingroup source was secondary to the encoding of the episode itself, then why source memory for ingroup trials were found to be better following correct than incorrect AC inferences? Shouldn't source memory for ingroup trials be generally low regardless of AC inference success? All in all, it was difficult to follow the authors' discussion of the source memory results, and I suggest that the authors to elaborate more on their discussion of the differing ingroup vs. outgroup source memory pattern and how this difference may relate to differential inferential success.

On p. 20, the authors suggest that “The enhanced encoding of ingroup information (e.g., Jeon et al., 2020; Marsh, 2020) may have boosted” both encoding- and retrieval-related mechanisms underlying successful inferences. This suggestion raises the question of whether the ingroup advantage observed for memory for AB associations contributed to the ingroup advantage in associative AC inference. I recommend that the authors incorporate the results from relevant analyses addressing this question in their revision, in the main text if space allows.

There were a few errors:

- “Outgroup source memory was higher for unsuccessful inferences” (on p. 20): Outgroup source memory was not significantly affected by inferential success, so this sentence was hard to follow. Perhaps, the authors wanted to say that outgroup source memory was better than ingroup source memory following unsuccessful inferences? Please clarify.
- “Even though the ABs were not directed associated with a social identity, these associations can come to mind during BC encoding [...]” (on p. 26): It is difficult to understand what the authors are saying here. Did I miss anything? Weren't all ABs presented with an ingroup or outgroup persona?

Response Letter to Reviewers

Reviewer 1

The study investigated how memory for object-scene pairs (AB) is affected by ingroup/outgroup status of a “persona” presented alongside them, as well as how the ingroup/outgroup status affects the likelihood of episodic inference that connects those AB associations with overlapping BC associations (AB & BC, therefore AC). The results showed that participants better remember object-scene AB associations and better inferred AC relationships when the AB associations were “presented” by an ingroup persona than outgroup persona. There was a flip side to this effect as the source memory (memory who presented a given object) was worse for ingroup than outgroup source. These findings provide an interesting connection between previously separate lines of research on episodic inference with research on source identity memory effects, and allude to how our biases towards information from ingroup sources may affect not only memory for individual information but also our tendency to connect related memories in service of knowledge formation.

This was a refreshing paper to read and a review to write. There was a lot to like. The studies were well designed, two studies were pre-registered, the authors self-replicated Study 1 in Study 3 after not finding an effect with a different manipulation in Study 2. The analyses were appropriate, results were interesting, the conclusions were well supported and carefully worded. Finally, the whole paper was very well written. It seems like a lot of thought went into each analysis decision as well as how much and where to report (e.g., adding Bayesian statistics when appropriate, including follow up exploratory analyses to corroborate some of the results).

We thank the reviewer for the encouraging evaluation of our manuscript.

I only have a few points I would like to see addressed.

1. Team source memory? One important set of findings related to team source memory, but it’s not possible to evaluate those as written. There needs to be more clarity and details for the methods, especially how it’s even possible to get the “team” correct but persona incorrect in a 2-alternative forced-choice test. As written and visualized in Fig 1, there were only two personas to choose from on every trial. Presumably, one of them was correct but the nature of the foil is not explained. If it’s always a persona from another team, then it wouldn’t be possible to choose a wrong persona from a correct team. If the foil can be from the same team as the correct choice (on some percentage of trials?), then essentially both answers are correct when analyzing for “team” source correct score, so it’s again not possible to compare 100% sure correct options across conditions, unless “?” (don’t know) responses are counted as incorrect on those trials. Either way, this needs to be much clearer.

We thank the reviewer for drawing our attention to this important issue and the need for a more detailed description. We have now clarified the methods of both Study 1 (p. 26) and Study 2 (p. 30). In the memory source test of Study 1 and 3, participants were presented with all four personas and were asked to select the accurate one. This setup enables differentiation between persona-based and team-based source memory. In Study 2, however, with only two personas presented, discerning between source memory for the team and persona it is not possible. We have updated Figure 1, which now displays the procedure of Study 1 and 3, where participants had to identify the source among all four personas.

2. AB effect driving AC effect? With one exception, all AC differences were accompanied by AB differences. It would be important to know if the AB differences fully account for AC differences or if AC differences go above and beyond AB differences. Are people unable to make AC inference because

they don't remember the AB trials? Or are there additional inference effects even after AB's were taken into account. There is probably many ways how to address this question. For instance, Zeithamova and Preston 2017 did two control analyses when faced with the same issue of AC differences being potentially driven by AB differences, one looking at across-subjects correlations, one limiting AC analysis to only those trials where AB was remembered. Just to clarify, the paper is still a worthwhile reading, even if it turns out that the AC differences can be fully explained by AB differences.

We concur with the reviewer that this is a valuable point to address. Following the suggestion, we tested if the ingroup advantage observed in ABs predicted the ingroup advantage observed in the ACs in Studies 1 and 3. For completeness, we also tested if the ingroup advantage observed in BCs (in Study 3) could predict the ingroup advantage in the ACs. Interestingly, we found that the ingroup advantage observed in direct associations did not predict indirect associations, suggesting that the inference ingroup advantage is not merely a function of the previously reported direct ingroup advantages. These analyses and findings have been incorporated into the main manuscript (see p. 9 and p. 16f), with a further discussion of the matter on p. 21.

3. In the discussion, the authors provide an example that concludes: "Our results suggest that you would be less likely to attribute this to the fundraiser of the disliked group than if your own group had been responsible." I was confused about this claim and did not find it consistent with the data. All the source memory effects (for team) were driven by lower Ingroup Incorrect Inference trials, with the remaining conditions (outgroup correct and incorrect AC, ingroup correct AC) being all about equal. Either revise the example or provide a more thorough handholding through the logic of your claim.

We appreciate the reviewer's comment and have revised the example in the text. The example aims to illustrate that we are more likely to make inferences across event boundaries if the information is provided by an ingroup source. Our findings suggest that we are more inclined to attribute the cleanliness (C) of the park (B) to the fundraiser (A) if arranged by an organization we like or are part of compared with an organization we dislike (p. 19f.). We believe that the revised example provides a clearer reflection of the associative inference task and more seamlessly illustrates the potential implications of our findings.

4. Detail memory. Detail memory was delegated to the supplement because no effects were found, which is fine. However, given the Carpenter & Schacter vs. de Araujo Sanches & Zeithamova conflicting findings, the null effect may be actually interesting. I would definitely like to see more information, even if it remains in the supplement. At minimum, please report the actual data with the actual error magnitude for each condition (not just anova results). I also didn't understand the scaling to values between 0 and 1 and how that would help with large variability, unless it's a nonlinear conversion. Nevertheless, using analyses on each subject's medians instead of means seemed like an appropriate choice given that the error distributions are necessarily very skewed.

We have added descriptive values for detail memory for each condition and study to Supplementary Note 4. These can be consulted in Supplementary Table 4.1. We have also clarified the scaling. The problem of high variance was only tackled by using medians for each participant, while the scaling was merely applied to make values more easily interpretable. The raw values were in screen size units from Pavlovia and higher values indicate a bigger difference between actual and indicated object position. We therefore scaled and flipped the values so that values of 1 would indicate the best detail memory (p. 9 of Supplement).

Minor comments:

- Fig 1 legend: "In Study 3, the items were assessed ..". Maybe switch to "the personas" to avoid confusion with objects?

We have clarified this ambiguous formulation in the legend of Figure 1 (p. 7).

- Why were the subjects US-Americans when the research team isn't?

Political orientation was the first attribute that participants had to assign to the personas, providing a sort of anchoring for the rest of the profile construction. Sweden, where our research team is located, is a multi-party democracy with eight parties currently represented in parliament. This would have made the group manipulation, where participants assigned party membership to the personas, less clear-cut, as differences between most parties are generally less prominent. We therefore decided to use the US, that has a two-party system with strong polarization (Huddy et al., 2018), to gather traction in the group manipulation. As this is a question that may be interesting for readers, we have added some information on this issue in the Participants section of Study 1 (p. 22).

- Provide age range in addition to the mean age for participants in all studies.

We thank the reviewer for noticing that this was absent from the text. This information is now added (pp. 22, 29, 30).

Reviewer 2

This manuscript investigated the effects of ingroup vs. outgroup status of information source on memory for inferential associations in three studies. In all studies, participants were first shown an object picture on a background image (AB) that was presented by a persona whose views/interests were similar vs. dissimilar to their own (ingroup vs. outgroup), and then a different object picture on the same background image (BC). Later, participants' memory for indirect, associative inferences (AC), direct associations (AB, BC), as well as for the information source (source memory: whether an object was presented by ingroup or outgroup) and the location (detail memory) of the individual objects were probed. Studies 1 and 3 used two personas per group, whereas Study 2 had only one persona per group. In Studies 1 and 2, interpersonal liking as a manipulation check was measured once on a relative scale (with the ingroup at one end of the scale and the outgroup at the other), whereas in Study 3 it was measured separately per each group. In all studies, participants liked ingroup personas more than outgroup personas. Also consistent across the studies, memory was better for direct AB, BC associations than for associative AC associations. Of direct relevance to this manuscript's inquiry, in Studies 1 and 3, there was an ingroup advantage in associative AC inference (as well as in memory for AB and BC associations). Also, in Study 1, source memory was better following successful than unsuccessful associative inference, but only for ingroup trials but not for outgroup trials, with outgroup source memory significantly better than ingroup source memory following successful inference. These source memory results were largely replicated in Experiment 3, with a caveat that some of the comparisons did not reach statistical significance. In comparison to these results from Studies 1 and 3, in Study 2, the authors found no significant effect of group on associative AC inference, direct AB and BC associations, or source memory.

This is an interesting paper that addresses an empirical question of how social group membership influences memory integration. It is a nice extension of previous studies on the effects of social group membership on memory processes, and the findings of this study will be informative for future studies aimed at investigating specific proximal mechanisms through which the perception of ingroup vs. outgroup affects everyday memory processes. The studies appear to have been competently designed and conducted, the statistical analyses are appropriate to the research

question, and the findings are clearly reported. Overall, I found the manuscript well-organized and well-written. However, I have a number of suggestions for improving the manuscript that I list below, roughly in the order they appear in the manuscript.

We thank the reviewer for these positive comments about our studies and manuscript.

The authors describe that their sample size for Study 1 was based on the sample size in previous studies but these studies they mention appear to bear no relevance to the critical ingroup vs. outgroup manipulation. Furthermore, the authors make no mention at all of power or expected effect sizes. Did the authors consult any past studies that looked at the effects of social group on memory processes (e.g., Jeon et al., 2021; Marsh, 2020 that the authors themselves cite in the manuscript)? Was the sample size of Study 1 sufficient to detect ingroup-related memory effects reported in such studies? In fact, given that this manuscript is concerned with memory for indirect, inferential associations rather than direct associations, the authors should expect their effect sizes to be somewhat smaller than the effect sizes reported in previous memory studies. In any case, I suggest that the authors offer an improved rationale for the sample size for Study 1.

We thank the reviewer for this observation. We also consulted studies that evaluated the effect of social group on memory. These studies typically have samples sizes of around 50 (Jeon et al., 2021; Xia et al., 2019) and they often use other paradigms and designs, such as social contagion paradigms with only a handful of trials (Andrews & Rapp, 2014), or between-subject designs (Marsh, 2020). Therefore, we believe that studies investigating associative inference mirrored our design more closely (Carpenter & Schacter, 2017). However, in agreement with the reviewer suggestion, we now provide an elaborated rationale for our sample size that also refers to studies investigating effects of social group on memory (p. 22). Given that this study was the very first one investigating this effect, we decided to be conservative and open 60 slots, also not knowing how big the drop out would be in the online setting.

There appears to be some room for improvement in terms of describing questions motivating Studies 2 and 3. For example, in Study 2, the authors briefly mention that they wanted to see if it is liked individuals vs. ingroup that gives rises to the ingroup advantage in associative inference, but provide no description at all about relevant background literature and/or hypotheses.

We thank the reviewer for this comment and have expanded the introduction of Study 2 to clarify our reasoning (p. 14). The attempted replication with one persona per group was based on the social memory literature, which often tests the effect not only with ingroup members within an explicit group framework, but also with liked, trusted, or known individuals, often finding similar effects. We therefore decided to test if the social effects in inferential memory would also generalize to liked individuals (instead of groups).

Likewise, in Study 3, the authors describe that they chose to measure interpersonal liking separately for each group (rather than once using a relative scale as in Studies 1 and 2) in order to examine a potential correlation between the ingroup advantage in associative inference and ingroup liking, but to me it is not very clear why the relative scale could not be used to test for the correlation. I suggest the authors to further elaborate on the motivation/rationale for each study to improve their already well-written manuscript even further.

We are grateful to the reviewer for this very relevant comment. The rationale for the use of separate liking scales has now been clarified in the introduction of Study 3 (p. 15f). The use of the separate scales made it possible to obtain independent measures for ingroup liking and outgroup disliking, creating alignment with the conventional use of liking measures like the IL-6 that we used (Veksler &

Eden, 2017). First, the separate scales improved the group manipulation check, as we could now analyze not only relative ingroup preference but also whether the manipulation induced both ingroup like and outgroup dislike. Second, we expected this assessment to be less taxing for participants as they did not have to compare between personas while making judgements. We leveraged this improved assessment to test whether the AC ingroup advantage would co-vary with the liking ingroup bias in order to bolster the connection between the social group manipulation and the observed inference effect.

Across both the main text and the supplementary material, there are a good number of analyses that the authors performed, but without providing clear justification (e.g., categorizing trials based on the ease of encoding for source memory analysis, partial source memory analysis). For example, it is not very clear why the authors decided to look at partial source memory, and how the results of partial source memory relate to ingroup vs. outgroup inferential success and/or to the absence of ingroup advantage in associative inference when there is only one persona (in Study 2). I suggest the authors to provide a clearer justification/explanation for why they made certain analytical choices and discuss the results of different analyses in light of the main inquiry of the manuscript. This will greatly help the reader to more clearly follow the results from each separate analysis and to better understand their implications.

We thank the reviewer for noticing that some aspects of the analysis lacked a clear rationale. First, we have now clarified the categorization of the trials for the analysis (see Supplementary Note 2, p. 4). Second, we have also provided the rationale for the partial source memory analysis (p.26f of the manuscript and Supplementary Note 5 p. 11). This analysis was conducted to check our group manipulation. Previous studies have showed that outgroup is associated with lower individualization (Brewer et al., 1995; Crump et al., 2010). Thus, we tested if this was evident also in our experiment, using the partial source memory indicator. This was the case, supporting the success of the manipulation. To make this clearer we now present this analysis together with the manipulation checks.

For the sake of completeness, I would like to ask the authors to report Bayes factors (i.e., “amount of evidence”) for not only null effects but also significant effects consistently throughout the manuscript. On a related note, I suggest the authors to steer away from calling non-significant results as “marginal” and provide Bayes factors for what they deemed “marginal”. Doing so will allow the reader to gauge the amount of evidence for the reported non-significant results (and any follow-up planned comparisons that were done despite the non-significant null effect).

We thank the reviewer for this comment. We have removed all references to “marginal effects”. Instead, Bayes factors are presented for all the effects that were deemed “marginal”.

In the discussion, the authors state “The higher source monitoring resources allocated to the outgroup may interfere with encoding the episode itself, leading to lowered mnemonic flexibility for outgroup information. This is different for trustworthy ingroup information where the encoding of the source may be secondary to the episode itself.” (on p. 20). To me this is difficult to follow: if what the authors suggest is the case, shouldn’t source memory for outgroup trials better following incorrect than correct AC inferences (i.e., incorrect AC inferences resulting from even greater interference with encoding the episode itself due to outgroup source monitoring)? Likewise, if encoding of the ingroup source was secondary to the encoding of the episode itself, then why source memory for ingroup trials were found to be better following correct than incorrect AC inferences? Shouldn’t source memory for ingroup trials be generally low regardless of AC inference success? All in all, it was difficult to follow the authors’ discussion of the source memory results, and I suggest

that the authors to elaborate more on their discussion of the differing ingroup vs. outgroup source memory pattern and how this difference may relate to differential inferential success.

We appreciate that the reviewer highlights that the discussion concerning source memory patterns across ingroup and outgroup was unclear. The interpretation of our findings is focused on situations when encoding is difficult, in which participants need to prioritize which information to encode. When encoding is difficult, the ingroup sources may be sacrificed for episode information, while outgroup sources take dominance over episodic information. Under easy encoding conditions, it is more likely that the whole episode is encoded and no differences between ingroup and outgroup source are expected to be observed.

This interpretation is corroborated by the analysis presented in the Supplementary Note 2. Here, we show that difficult encoding circumstances that still lead to correct AC inferences are associated with a loss of source information for the ingroup. However, the source of outgroup information is kept, even under difficult encoding circumstances. We have now substantially revised the text about the source memory in the discussion of Study 1 (p. 11) and in the main discussion (p. 20), making it clearer that the interpretation is focused on variation in encoding difficulty.

On p. 20, the authors suggest that “The enhanced encoding of ingroup information (e.g., Jeon et al., 2020; Marsh, 2020) may have boosted” both encoding- and retrieval-related mechanisms underlying successful inferences. This suggestion raises the question of whether the ingroup advantage observed for memory for AB associations contributed to the ingroup advantage in associative AC inference. I recommend that the authors incorporate the results from relevant analyses addressing this question in their revision, in the main text if space allows.

We thank the reviewer for this valuable suggestion. We ran the proposed analyses and found no evidence that the ingroup biases effect found in the AB/BC direct associations predicts the AC indirect inference. These analyses and results have been incorporated into the text (see p. 9 and p. 16f). See also comment to Reviewer 1.

There were a few errors:

- ***“Outgroup source memory was higher for unsuccessful inferences” (on p. 20): Outgroup source memory was not significantly affected by inferential success, so this sentence was hard to follow. Perhaps, the authors wanted to say that outgroup source memory was better than ingroup source memory following unsuccessful inferences? Please clarify.***

We thank the reviewer for spotting this mistake. They are absolutely right, and this sentence was missing the qualification that it was higher than the ingroup for unsuccessful inferences. We have corrected this error (p. 20).

- ***“Even though the ABs were not directly associated with a social identity, these associations can come to mind during BC encoding [...]” (on p. 26): It is difficult to understand what the authors are saying here. Did I miss anything? Weren’t all ABs presented with an ingroup or outgroup persona?***

Once again, we thank the reviewer for pointing out this error – it is indeed the BCs that were not directly associated with a social identity, not the ABs (now on p. 27).

References

- Andrews, J. J., & Rapp, D. N. (2014). Partner characteristics and social contagion: Does group composition matter? *Applied Cognitive Psychology, 28*(4), 505–517. <https://doi.org/10.1002/acp.3024>
- Brewer, M. B., Weber, J. G., & Carini, B. (1995). Person memory in intergroup contexts: Categorization versus individuation. *Journal of Personality and Social Psychology, 69*(1), 29–40. <https://doi.org/10.1037/0022-3514.69.1.29>
- Carpenter, A. C., & Schacter, D. L. (2017). Flexible retrieval: When true inferences produce false memories. *Journal of Experimental Psychology: Learning, Memory, and Cognition, 43*(3), 335–349. <https://doi.org/10.1037/xlm0000340>
- Crump, S. A., Hamilton, D. L., Sherman, S. J., Lickel, B., & Thakkar, V. (2010). Group entitativity and similarity: Their differing patterns in perceptions of groups. *European Journal of Social Psychology, 40*(7), 1212–1230. <https://doi.org/10.1002/ejsp.716>
- Huddy, L., Bankert, A., & Davies, C. (2018). Expressive versus instrumental partisanship in multiparty European systems. *Political Psychology, 39*(S1), 173–199. <https://doi.org/10.1111/pops.12482>
- Jeon, Y. A., Banquer, A. M., Navangul, A. S., & Kim, K. (2021). Social group membership and an incidental ingroup-memory advantage. *Quarterly Journal of Experimental Psychology, 74*(1), 166–178. <https://doi.org/10.1177/1747021820948721>
- Marsh, B. U. (2020). The cost of racial salience on face memory: How the cross-race effect is moderated by racial ambiguity and the race of the perceiver and the perceived. *Journal of Applied Research in Memory and Cognition, 10*(1), 13–23. <https://doi.org/10.1037/h0101790>
- Veksler, A. E., & Eden, J. (2017). Measuring interpersonal liking as a cognitive evaluation: Development and validation of the IL-6. *Western Journal of Communication, 81*(5), 1–16. <https://doi.org/10.1080/10570314.2017.1309452>
- Xia, R., Su, W., Wang, F., Li, S., Zhou, A., & Lyu, D. (2019). The moderation effect of self-enhancement on the group-reference effect. *Frontiers in Psychology, 10*, Article 1463. <https://doi.org/10.3389/fpsyg.2019.01463>

2nd Nov 23

Dear Dr Bramao,

Your manuscript titled "Inferential memory is moderated by the ingroup/outgroup status of the information source" has now been seen by our reviewers, whose comments appear below. In light of their advice I am delighted to say that we are happy, in principle, to publish a suitably revised version in *Communications Psychology* under the open access CC BY license (Creative Commons Attribution v4.0 International License).

We therefore invite you to revise your paper one last time to address the remaining concerns of our reviewers and a list of editorial requests. At the same time we ask that you edit your manuscript to comply with our format requirements and to maximise the accessibility and therefore the impact of your work.

Please note that it may still be possible for your paper to be published before the end of 2023, but in order to do this we will need you to address these points as quickly as possible so that we can move forward with your paper.

EDITORIAL REQUESTS:

Please review our specific editorial comments and requests regarding your manuscript in the attached "Editorial Requests Table". Please outline your response to each request in the right-hand column. Please upload the completed table with your manuscript files as a Related Manuscript file.

I highlight in particular 3 requests:

1) The research involves US participants and was conducted on the basis of the Swedish Act concerning the Ethical Review of Research involving Humans (2003:460) and the World Medical Association's Declaration of Helsinki's Code of Ethics. Please include an ethics & inclusion statement (<https://www.nature.com/nature-portfolio/editorial-policies/authorship#authorship-inclusion-and-ethics-in-global-research>) in the manuscript, commenting in particular on questions 1 and 5.

2) *Communications Psychology* mandates the sharing of computational analysis code. Please review our requirements and specifications as detailed in the checklist and on our policy pages, prepare the code accordingly, and update the Code Availability Statement.

3) Please revise the results reporting throughout your manuscript. This request pertains to where results are reported (all main results should be in the manuscript, rather than the SI) and how results are reported (complete statistics are required, including for non-significant results). Please note that you cannot include new results in the Discussion. The reference to the preliminary study should be removed, as readers cannot ascertain its quality and the results are below conventional levels for moderate evidence.

SUBMISSION INFORMATION:

OPEN ACCESS:

Communications Psychology is a fully open access journal. Articles are made freely accessible on publication under a [CC BY license](http://creativecommons.org/licenses/by/4.0) (Creative Commons Attribution 4.0 International License). This license allows maximum dissemination and re-use of open access materials and is preferred by many research funding bodies.

For further information about article processing charges, open access funding, and advice and support from Nature Research, please visit <https://www.nature.com/commspsychol/article-processing-charges>

At acceptance, you will be provided with instructions for completing this CC BY license on behalf of all authors. This grants us the necessary permissions to publish your paper. Additionally, you will be asked to declare that all required third party permissions have been obtained, and to provide billing information in order to pay the article-processing charge (APC).

* DATA AVAILABILITY:

Please use the following link to submit the above items:
[link redacted]

Best regards,

Marike, on behalf of Jesse Rissman

Marike Schiffer, PhD
Chief Editor
Communications Psychology

REVIEWERS' COMMENTS:

Reviewer #1 (Remarks to the Author):

I have no further comments and find this to be a well-written and interesting paper.

Reviewer #2 (Remarks to the Author):

The authors have satisfactorily addressed my concerns with their revision. I now believe this manuscript to be a worthwhile scientific contribution suitable for publication. I congratulate the authors on their interesting research.

Response Letter to Reviewers and Editors

Editorial Requests

1) The research involves US participants and was conducted on the basis of the Swedish Act concerning the Ethical Review of Research involving Humans (2003:460) and the World Medical Association's Declaration of Helsinki's Code of Ethics. Please include an ethics & inclusion statement (<https://www.nature.com/nature-portfolio/editorial-policies/authorship#authorship-inclusion-and-ethics-in-global-research>) in the manuscript, commenting in particular on questions 1 and 5.

We have included in the methods section an ethics and inclusion statement as a separate heading.

2) Communications Psychology mandates the sharing of computational analysis code. Please review our requirements and specifications as detailed in the checklist and on our policy pages, prepare the code accordingly, and update the Code Availability Statement.

The code used is available and the code availability statement is updated.

3) Please revise the results reporting throughout your manuscript. This request pertains to where results are reported (all main results should be in the manuscript, rather than the SI) and how results are reported (complete statistics are required, including for non-significant results). Please note that you cannot include new results in the Discussion. The reference to the preliminary study should be removed, as readers cannot ascertain its quality and the results are below conventional levels for moderate evidence.

We have revised the results according to the recommendations: 1) All main results are reported in the manuscript; 2) all the statistics are reported (including the non-significant findings); and 3) the preliminary study is removed from the manuscript and supplement.

Reviewer 1

I have no further comments and find this to be a well-written and interesting paper.

We thank the reviewer for their time and effort in the review and their helpful comments in the process.

Reviewer 2

The authors have satisfactorily addressed my concerns with their revision. I now believe this manuscript to be a worthwhile scientific contribution suitable for publication. I congratulate the authors on their interesting research.

We are glad to have addressed the helpful suggestions and questions raised in the review and we thank the reviewer for their time and effort.